# Meiosis I chromosome segregation is established through regulation of microtubule–kinetochore interactions

**Matthew P Miller[1][†], Elçin Ünal[1][†], Gloria A Brar[2], Angelika Amon[1]\***

[1]Department of Biology, Massachusetts Institute of Technology, Cambridge, United States; [2]Department of Cellular and Molecular Pharmacology, University of California, San Francisco, San Francisco, United States

**Abstract** During meiosis, a single round of DNA replication is followed by two consecutive rounds of nuclear divisions called meiosis I and meiosis II. In meiosis I, homologous chromosomes segregate, while sister chromatids remain together. Determining how this unusual chromosome segregation behavior is established is central to understanding germ cell development. Here we show that preventing microtubule–kinetochore interactions during premeiotic S phase and prophase I is essential for establishing the meiosis I chromosome segregation pattern. Premature interactions of kinetochores with microtubules transform meiosis I into a mitosis-like division by disrupting two key meiosis I events: coorientation of sister kinetochores and protection of centromeric cohesin removal from chromosomes. Furthermore we find that restricting outer kinetochore assembly contributes to preventing premature engagement of microtubules with kinetochores. We propose that inhibition of microtubule–kinetochore interactions during premeiotic S phase and prophase I is central to establishing the unique meiosis I chromosome segregation pattern.

**\*For correspondence:**
angelika@mit.edu

[†]These authors contributed equally to this work

**Competing interests:** The authors have declared that no competing interests exist

## Introduction

Cells have evolved intricate mechanisms to execute proper partitioning of the genetic material during cell division. This task is especially complex in meiosis, the cell division used by sexually reproducing organisms to generate gametes. The goal of meiosis is to reduce the genome content by half such that proper ploidy is maintained upon fusion of gametes. To achieve this, a single round of DNA replication is followed by two consecutive rounds of nuclear division called meiosis I and meiosis II. During meiosis I homologous chromosomes segregate. Meiosis II resembles mitosis in that sister chromatids segregate from each other. The establishment of this specialized chromosome segregation pattern requires three changes that modulate how chromosomes interact with each other and with the microtubule cytoskeleton: (1) reciprocal recombination between homologous chromosomes, (2) the way linkages between sister chromatids, known as sister-chromatid cohesion, are removed from chromosomes and (3) the manner in which chromosomes attach to the meiotic spindle.

Homologous recombination is initiated by programmed double-strand breaks (DSBs), which are catalyzed by Spo11 following premeiotic DNA replication (*Keeney et al., 1997*). Subsequent repair of DSBs by crossover recombination generates physical linkages between homologous chromosomes. This, in turn, allows homologs to attach to the meiosis I spindle such that each homolog interacts with microtubules emanating from opposite spindle poles. As a result, homologous chromosomes biorient on the meiosis I spindle. The spindle assembly checkpoint prevents the onset of chromosome segregation until this process is completed. Once each pair of homologs is bioriented, checkpoint signaling ceases and anaphase entry ensues. A ubiquitin ligase known as the anaphase promoting complex/cyclosome

**eLife digest** Diploid organisms contain two sets of chromosomes, one set inherited from the mother and the other from the father. Humans, for example, have 23 pairs of chromosomes, and the chromosomes within each pair are said to be homologous because they are similar to each other in a number of ways, including length and shape. When it comes time for one of these cells to duplicate, each chromosome is first replicated to generate a pair of identical chromosomes called sister chromatids, which subsequently separate in a cell division process known as mitosis to produce two identical daughter cells.

While most cells proliferate via mitotic cell division, the germ cells that generate gametes in the form of sperm or eggs undergo a different cell division known as meiosis. This process reduces the number of chromosomes by a factor of two, so that the original number of chromosomes is restored by the fusion of gametes during sexual reproduction. During meiotic cell division, a single round of DNA replication is followed by two consecutive rounds of nuclear division called meiosis I and meiosis II. During meiosis I, homologous chromosomes are separated. Subsequently, during meiosis II, the sister chromatids separate to produce a total of four products, each with half the number of chromosomes as the original cell.

The separation of homologous chromosomes or sister chromatids relies on them being pulled apart by microtubules. One end of each microtubule is attached to a protein-based structure called a kinetochore, which is assembled onto the centromere of each chromosome. The other end of each microtubule is attached to a structure that is called a centrosome in human cells and a spindle pole body in yeast cells. Human cells have two centrosomes, which reside on the opposite poles of the cell, and likewise for the spindle pole bodies in yeast cells. In mitotic cells and in meiosis II, microtubules attach to kinetochores in a way that means the sister chromatids are pulled apart. During meiosis I, on the other hand, they attach to kinetochores in a manner so the homologous chromosomes are pulled apart.

Miller et al. now show how the timing of the interaction between the kinetochore and microtubules is critical to ensure that the homologous chromosomes are separated during meiosis I. They found that premature interactions resulted in the separation of sister chromatids (as happens in mitosis) rather than the separation of homologous chromosomes, as is supposed to happen in meiosis I. They also showed that cells prevent such premature interactions by dismantling the outer regions of the kinetochore and reducing the levels of enzymes called CDKs in the cell. These results demonstrate that preventing premature microtubule–kinetochore interactions is essential for establishing a meiosis I-specific chromosome architecture, and they also provide fresh insights into how the molecular machinery that is responsible for mitotic chromosome segregation can be modulated to achieve meiosis.

and its specificity factor Cdc20 (APC/C-Cdc20) targets Securin for degradation, relieving Separase inhibition (*Cohen-Fix et al., 1996*; *Ciosk et al., 1998*). Separase is a protease that cleaves the kleisin subunit of cohesin, the protein complex that mediates sister-chromatid cohesion (*Uhlmann et al., 1999*, *2000*; *Schleiffer et al., 2003*). In meiosis I, cleavage of cohesin at chromosome arms allows homologs to segregate (*Buonomo et al., 2000*). However, cohesin around the centromeres is protected from cleavage during meiosis I, which is essential for the accurate segregation of sister chromatids during meiosis II. Protection of centromeric cohesin is accomplished by preventing phosphorylation of Rec8, the meiosis-specific kleisin. This occurs, at least in part, by Sgo1 (MEI-S332)-dependent recruitment of the protein phosphatase PP2A to centromeric regions where it antagonizes Rec8 phosphorylation (*Kerrebrock et al., 1995*; *Katis et al., 2004a*; *Kitajima et al., 2004*, *2006*; *Riedel et al., 2006*).

The third modification necessary to bring about the meiotic chromosome segregation pattern is the manner in which kinetochores attach to microtubules during meiosis I and meiosis II. In meiosis I, kinetochores of sister chromatid pairs (henceforth sister kinetochores) attach to microtubules emanating from the same spindle pole, a process called sister kinetochore coorientation. During meiosis II, as during mitosis, sister kinetochores attach to microtubules emanating from opposite spindle poles and are thus bioriented (reviewed in *Marston and Amon, 2004*). In budding yeast, sister kinetochore

coorientation is brought about by the monopolin complex, which consists of Mam1, Lrs4, Csm1 and the casein kinase 1, Hrr25 (*Toth et al., 2000*; *Rabitsch et al., 2003*; *Petronczki et al., 2006*). Lrs4 and Csm1 localize to the nucleolus during interphase. During exit from pachytene, a stage of prophase I, Lrs4 and Csm1 associate with Mam1 and Hrr25 at kinetochores, a process that requires the Polo kinase Cdc5 (*Clyne et al., 2003*; *Lee and Amon, 2003*; *Matos et al., 2008*). How the association of monopolin with kinetochores is coordinated with respect to kinetochore assembly and microtubule–kinetochore interactions during meiosis is not understood.

Cyclin-dependent kinases (CDKs) are the central regulators of the mitotic and meiotic divisions. In budding yeast, a single CDK associates with one of six B-type cyclins (Clb1-Clb6) (reviewed in *Morgan, 1997*). In meiosis, Clb5- and Clb6-CDKs drive DNA replication and recombination, whereas Clb1-, Clb3- and Clb4-CDKs promote the meiotic nuclear divisions (reviewed in *Marston and Amon, 2004*). Meiotic cyclin-CDK activity is regulated both at the transcriptional and translational level (*Grandin and Reed, 1993*; *Carlile and Amon, 2008*). Transcription of *CLB1*, *CLB3* and *CLB4* occurs only after exit from pachytene (*Chu and Herskowitz, 1998*); *CLB3* is also translationally repressed during meiosis I, thus restricting Clb3-CDK activity to meiosis II (*Carlile and Amon, 2008*). The major mitotic cyclin, *CLB2*, is not expressed during meiosis (*Grandin and Reed, 1993*).

Here we investigate the importance of cyclin-CDK regulation in establishing the meiotic chromosome segregation pattern. We show that expression of a subset of cyclins during premeiotic S phase and early prophase I, defined as the prophase stages up to exit from pachytene, causes premature microtubule–kinetochore interactions. This, in turn, disrupts both sister kinetochore coorientation and protection of centromeric cohesin during meiosis I, revealing that the temporal control of microtubule–kinetochore interactions is essential for meiosis I chromosome morphogenesis. Furthermore, we define the mechanism by which premature microtubule–kinetochore interactions are prevented; through regulation of cyclin-CDK activity and of outer kinetochore assembly. Our results demonstrate that preventing premature microtubule–kinetochore interactions is essential for establishing a meiosis I-specific chromosome architecture and provide critical insights into how the mitotic chromosome segregation machinery is modulated to achieve a meiosis I-specific pattern of chromosome segregation.

## Results

### Cyclin expression is sufficient to induce spindle formation and microtubule–kinetochore interactions

We previously reported that *CLB3* expression prior to meiosis I induces a change in the pattern of chromosome segregation such that sister chromatids, instead of homologous chromosomes, segregate during the first nuclear division (*Carlile and Amon, 2008*). To determine how Clb-CDKs impact meiotic chromosome segregation and whether Clb-CDKs play redundant or specific roles in regulating this process, we examined the consequences of prematurely expressing *CLB1*, *CLB3*, *CLB4* or *CLB5*.

In our previous studies we expressed *CLB3* from the *GAL1-10* promoter driven by an estrogen inducible Gal4-ER fusion (*Carlile and Amon, 2008*). Expression from the *GAL1-10* promoter led to Clb3 accumulation in meiosis I to levels that are comparable to those seen in meiosis II in wild-type cells (*Carlile and Amon, 2008*). However, estrogen interferes with meiotic progression when added during early stages of sporulation (*Figure 1A*). To circumvent this problem we utilized the copper-inducible *CUP1* promoter to drive Clb3 expression. Expression from the *CUP1* promoter led to approximately fivefold higher levels of Clb3 protein compared to expression from the *GAL1-10* promoter (*Figure 1B*). To examine the consequences of the two *CLB3* constructs on chromosome segregation we used *GAL-CLB3* and *CUP-CLB3* strains in which one of the two homologs of chromosome III was marked by integrating a tandem array of tetO sequences ~20 kb from CENIII (heterozygous LEU2-GFP dots). These cells also expressed a tetR-GFP fusion, which allowed visualization of the tetO arrays (*Michaelis et al., 1997*). The analysis of GFP dot segregation during the first meiotic division revealed that despite the difference in Clb3 protein levels, the extent of sister chromatid segregation in meiosis I was similar between *GAL-CLB3* and *CUP-CLB3* cells (*Figure 1C*). This finding indicates that expression of Clb3 from either the *CUP1* or *GAL1-10* promoter efficiently induces sister chromatid segregation during meiosis I. Furthermore, the timing of when Clb3 is expressed, rather than the amount of Clb3 present, appears to be the primary determinant of this phenotype. Based on this observation and the finding that all four cyclins showed equal expression when produced from the *CUP1* promoter (*Figure 1D*) we utilized the *CUP1* promoter for most subsequent analyses.

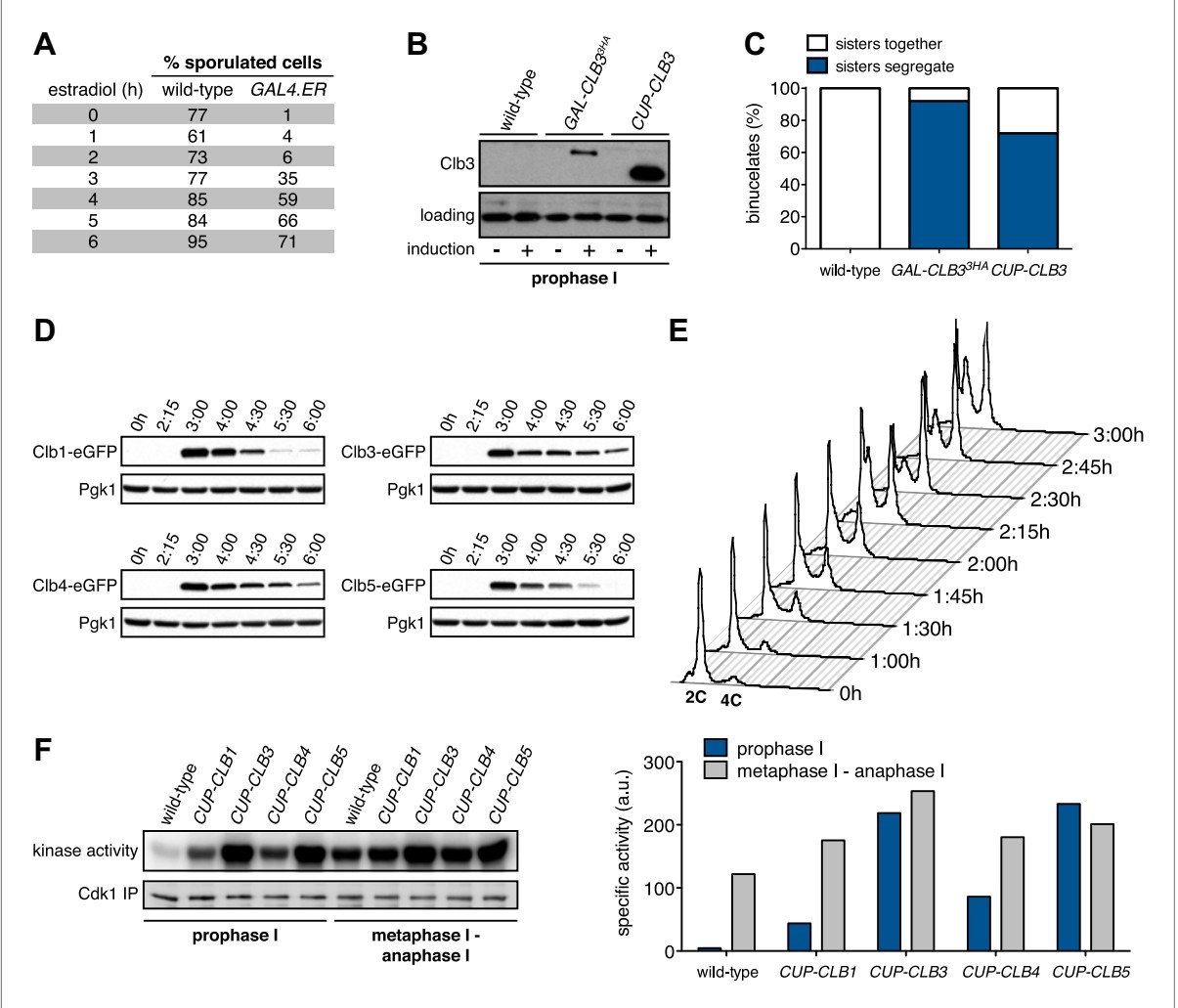

Figure 1. Characterization of premature cyclin expression and corresponding total CDK activity. (A) Wild-type (A4962) and *GAL4-ER* (A19151) cells were induced to sporulate. At the indicated time points, an aliquot was removed and treated with estradiol (1 µM). The percentage of cells that had sporulated after 24 hr was calculated as the sum of dyads, triads and tetrads divided by the total number of cells (n > 100 cells counted for each condition). (B) Wild-type (A18686), *GAL-CLB3-3HA* (A23084) and *CUP-CLB3* (A23086) cells also carrying the *GAL4-ER* fusion were induced to sporulate. After 3 hr, *CLB3* was induced. Each culture was treated with estradiol (1 µM) and $CuSO_4$ (50 µM). Cells were harvested after 1 hr of estradiol and $CuSO_4$ treatment for protein extraction. Levels of Clb3 were examined by Western blot analysis. A cross-reacting band was used as a loading reference. (C) Segregation of sister chromatids (equational division) using heterozygous GFP dots integrated at *LEU2* (~20 kb from CENIII) was quantified in binucleate cells from wild-type (A18686), *GAL-CLB3-3HA* (A23084) and *CUP-CLB3* (A23086). Note that the samples were collected from the same experiment described in (B) at a time point when a fraction of the cells had completed meiosis I (6 hr 30 min and 7 hr after induction of sporulation) (n > 100 for each sample). Using a chi-square test (df 1), the fraction of binucleates that display a reductional or equational division was compared between wild-type and *GAL-CLB3-3HA* $\chi^2$ = 166.4, p<0.0001 and between wild-type and *CUP-CLB3* $\chi^2$ = 108.7, p<0.0001. (D) Wild-type or *CUP-CLB-eGFP* cells also carrying the *GAL4-ER* and *GAL-NDT80* fusions were induced to sporulate. After 2 hr 15 min, cyclins were induced by addition of $CuSO_4$ (50 µM). Cells were released from the *NDT80* block at 4 hr 30 min post transfer to sporulation medium. Cyclin levels monitored by Western blot at the indicated time points in *CUP-CLB1-eGFP* (A28531), *CUP-CLB3-eGFP* (A28533), *CUP-CLB4-eGFP* (A28535) and *CUP-CLB5-eGFP* (A33199) cells. Pgk1 was used as a loading control. (E) Wild-type (A22678) cells carrying the *GAL4-ER* and *GAL-NDT80* fusions were induced to sporulate and $CuSO_4$ (50 µM) was added 2 hr 15 min after transfer into sporulation medium. Samples were taken at indicated time points to determine DNA content by flow cytometry. By 2 hr 15 min 43% of cells had a 4C DNA content. (F) Left: Wild-type (A28663), *CUP-CLB1* (A28665), *CUP-CLB3* (A28667), *CUP-CLB4* (A28669) and *CUP-CLB5* (A28671) cells carrying the *GAL4-ER* and *GAL-NDT80* fusions were induced to sporulate and $CuSO_4$ (50 µM) was added 2 hr 30 min after transfer into sporulation medium. In vitro kinase assays were performed with Cdc28-3V5 (Cdk1) immunoprecipitated from prophase I samples (collected 4 hr 30 min after sporulation induction, at the time of *NDT80* block-release) and metaphase I–anaphase I samples (collected 1 hr 30 min after release from the *NDT80* block). Right: specific activity was calculated by normalizing the amount of phosphorylated Histone H1 to the amount of immunoprecipitated Cdc28-3V5 using ImageQuant software (Molecular Dynamics, Sunnyvale, CA).

Having established a system to effectively express various cyclins prior to meiosis I we next examined the consequences of their premature expression on meiosis I events. We first asked whether misexpression of various cyclins is sufficient to induce spindle formation in cells arrested in pachytene of prophase I, due to lack of the transcription factor Ndt80 (*Xu et al., 1995*; *Chu and Herskowitz, 1998*). We induced cyclin expression from the *CUP1* promoter 135 min after the induction of sporulation when typically 40–65% of the cells have replicated their DNA (*Figure 1E*; *Blitzblau et al., 2012*) and examined spindle pole body (SPB, centrosome equivalent in budding yeast) separation and spindle morphology following induction. As expected, wild-type cells did not form spindles in the absence of *NDT80* function. Expression of *CLB5* from the *CUP1* promoter did not lead to SPB separation and spindle formation either, although expression of *CLB5* in the prophase I arrest led to a significant increase in total CDK activity (*Figures 1F and 2A*, *Figure 2—figure supplement 1*). In contrast, *CUP-CLB1*, *CUP-CLB3* and *CUP-CLB4* cells separated SPBs and formed bipolar spindles, shortly after copper addition (*Figure 2A* and *Figure 2—figure supplement 1*). Similar results were observed in cells with intact *NDT80* (data not shown). We conclude that expression of *CLB1*, *CLB3* or *CLB4* is sufficient to promote bipolar spindle assembly in *NDT80*-depleted cells.

Next, we determined whether expression of *CLB1*, *CLB3* or *CLB4* in pachytene-arrested cells also affects the manner in which chromosomes attach to the meiotic spindle using live-cell imaging. To this end we used strains carrying heterozygous CENV-GFP dots and an Spc42-mCherry fusion (Spc42 is an SPB component) to monitor the behavior of the marked centromere with respect to the spindle axis. In wild-type and *CUP-CLB5* cells, sister kinetochores remained closely associated with each other and did not appear to be tightly associated with SPBs, consistent with the observation that these cells failed to form a spindle. In contrast, we observed dynamic separation of heterozygous CENV-GFP dots upon expression of *CLB1* or *CLB3*, with sister kinetochores frequently splitting and coming together (*Figure 2B,C*). This observation is reminiscent of the behavior of bioriented sister chromatids during metaphase of mitosis (*Pearson et al., 2001*).

Cells expressing *CLB4* did not show transient splitting of sister kinetochores in prophase I, indicating that chromosomes are either unable to attach to the spindle or that homologous chromosomes, instead of sister chromatids, are bioriented as occurs in wild-type cells during metaphase I. To distinguish between these possibilities, we examined the behavior of *CUP-CLB4* cells in which both homologs of chromosome V harbor CENV-GFP dots (henceforth homozygous CENV-GFP dots). Similar to wild-type, we observed that in *CUP-CLB4* cells the two CENV-GFP dots remained tightly associated in prophase I, indicating that the homologous chromosomes are paired and not attached to the prematurely formed spindle (*Figure 2—figure supplement 2*). Together, these results indicate that *CUP-CLB1*, *CUP-CLB3* or *CUP-CLB4* expression promotes bipolar spindle formation in pachytene-arrested cells, but only *CLB1* and *CLB3* expression can promote stable microtubule–kinetochore attachments sufficient to generate tension.

To determine whether different amounts of total CDK activity were responsible for the phenotypic differences of prematurely expressing Clb1 or Clb3 compared to Clb4, we measured total CDK activity (Cdc28 in budding yeast) using Histone H1 as a substrate. Cdc28-associated kinase activity was low during prophase I and increased more than 25-fold during metaphase I/anaphase I in wild-type cells (*Figure 1F*). Expression of all four cyclins led to a significant increase in total CDK activity in prophase I (*Figure 1F*), but importantly, the degree of increase did not correlate with the ability to induce sister chromatid splitting in the *NDT80* arrest. For example, Clb1 expression led to a similar increase in Cdc28-associated kinase activity as expression of Clb4, yet Clb1 induced sister chromatid splitting whereas Clb4 did not (*Figures 1F and 2B,C*). We conclude that the ability to induce sister chromatid splitting does not correlate with total CDK activity produced by the various *CUP-CLB* fusions. Furthermore, SPB separation and spindle formation are not sufficient to induce microtubule–kinetochore interactions. Events that can be triggered by Clb1 and Clb3, but not Clb4 are also necessary to promote attachments sufficient to generate tension. Determining why *CLB4* expressing cells fail to form productive microtubule–kinetochore interactions could provide important insights into substrate specificity of cyclin-CDK complexes.

## Expression of *CLB3* or *CLB1* during premeiotic S phase/prophase I causes sister chromatids to segregate during meiosis I

To determine the consequences of premature cyclin expression on meiosis I chromosome segregation, we examined the segregation of heterozygous CENV-GFP dots in cells that were reversibly arrested in

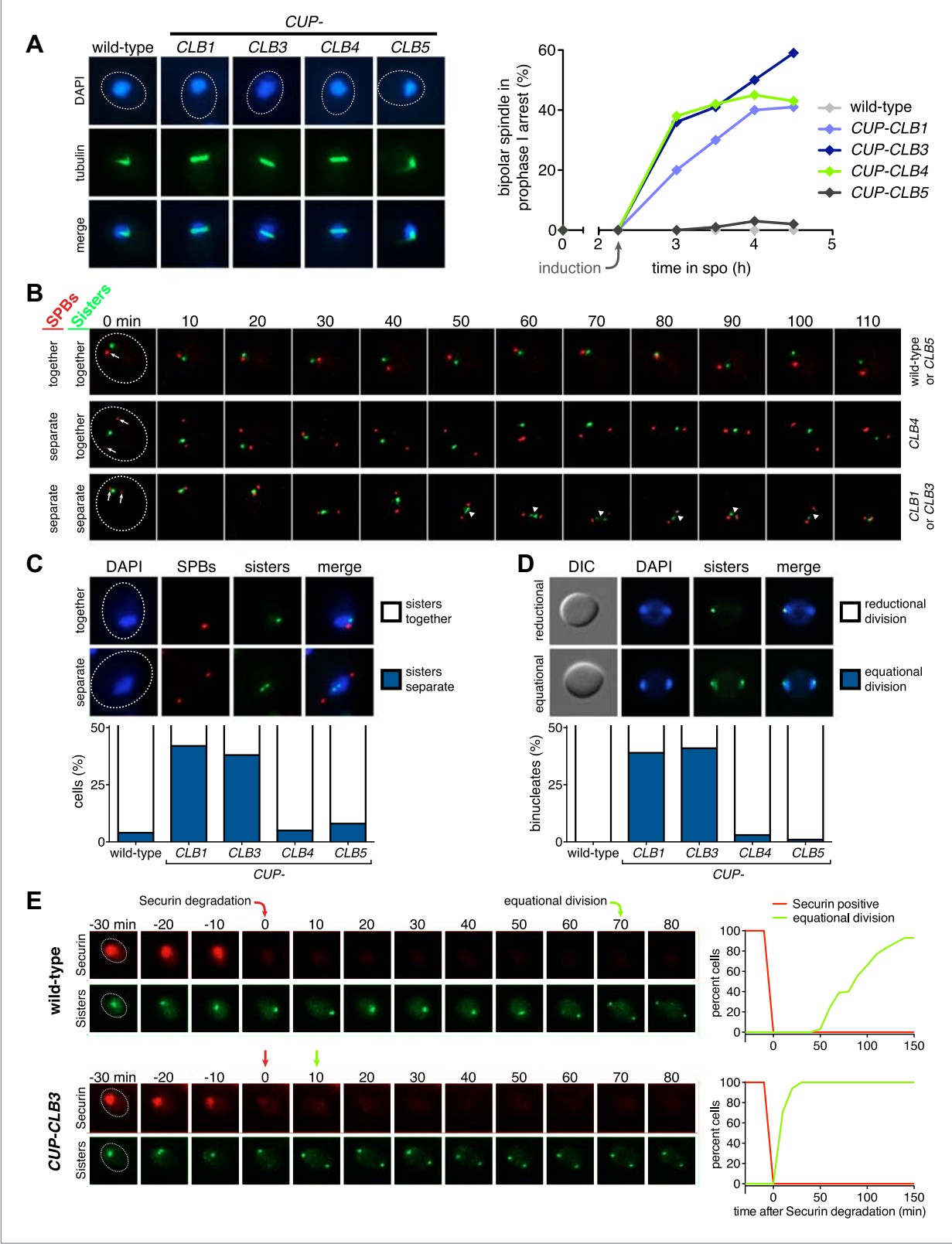

**Figure 2**. Premature expression of *CLB1* or *CLB3* causes sister kinetochore biorientation during prophase I and sister chromatid segregation in meiosis I. Wild-type or *CUP-CLB* cells were induced to sporulate. After 2 hr 15 min, cyclins were induced by addition of CuSO₄ (50 μM). Cells were either arrested during prophase I or released from an *NDT80* block 4 hr 30 min after induction of sporulation. (**A**) Bipolar spindle formation determined in wild-type (A22678),

*Figure 2. Continued on next page*

*Figure 2. Continued*

*CUP-CLB1* (A27421), *CUP-CLB3* (A22702), *CUP-CLB4* (A27423) and *CUP-CLB5* (A27425) during prophase I (n = 100 per time point). Images on left show spindle formation in *CUP-CLB* cells 4 hr after induction of sporulation; in this and all subsequent Figures microtubules are shown in green and DNA in blue. The dotted line depicts the cell membrane. (**B**) Microtubule–kinetochore engagement monitored during prophase I, starting at 1 hr after CuSO$_4$ addition in wild-type (A30700), *CUP-CLB1* (A30702), *CUP-CLB3* (A30704), *CUP-CLB4* (A30707) and *CUP-CLB5* (A30708) by live cell microscopy. SPBs (marked by arrow) and heterozygous CENV-GFP dots are shown (arrowheads mark separated CENV dots). In this and all subsequent figures SPBs are in red, GFP dots are in green. (**C**) Top panel: representative images of wild-type (A30700) and *CUP-CLB3* (A30704). Bottom panel: separation of hetero-zygous CENV-GFP dots in prophase I-arrested cells quantified in wild-type (A22678), *CUP-CLB1* (A27421), *CUP-CLB3* (A22702), *CUP-CLB4* (A27423) and *CUP-CLB5* (A27425) by live cell microscopy (over the duration of 8 hr, n > 100) as described in the 'Materials and methods'. The fraction of nuclei that display sister kinetochores as separate or together for each *CUP-CLB* strain was compared to wild-type using a chi-square test (df 1): *CUP-CLB1*, $\chi^2$ = 40.77, p<0.0001; *CUP-CLB3*, $\chi^2$ = 34.84, p<0.0001; *CUP-CLB4*, $\chi^2$ = 0.1163, p=0.7330; *CUP-CLB5*, $\chi^2$ = 1.418, p=0.2337. (**D**) Segregation of sister chromatids (equational division) using heterozygous CENV-GFP dots quantified in binucleates from wild-type (A22678), *CUP-CLB1* (A27421), *CUP-CLB3* (A22702), *CUP-CLB4* (A27423) and *CUP-CLB5* (A27425) (n = 100). The fraction of binucleates that display a reductional or equational division for each *CUP-CLB* strain was compared to wild-type using a chi-square test (df 1): *CUP-CLB1*, $\chi^2$ = 45.13, p<0.0001; *CUP-CLB3*, $\chi^2$ = 48.22, p<0.0001; *CUP-CLB4*, $\chi^2$ = 1.020, p=0.3124; *CUP-CLB5*, $\chi^2$ = 0, p=1. (**E**) Wild-type (A31019) and *CUP-CLB3* (A31021) cells monitored for segregation of heterozygous CENV-GFP dots with respect to Pds1 (Securin, red) degradation by live cell microscopy (n > 17). Time of Pds1 degradation set to t = 0, percent cells were plotted as a Kaplan–Meier curve. Note that for A31021, the analysis of cells that segregate sister chromatids in the first nuclear division is shown. Pds1 accumulation during prophase II is not observed using the Pds1-tdTomato construct, likely due to delayed maturation of the fluorophore (***Katis et al., 2010***).

The following figure supplements are available for figure 2.

**Figure supplement 1**. Spindle pole body separation in *CUP-CLB* cells.

**Figure supplement 2**. Homolog separation in *CUP-CLB4* cells.

**Figure supplement 3**. Chromosome III sister chromatid segregation in *CUP-CLB3* cells.

**Figure supplement 4**. Sister chromatid segregation in *CUP-CLB3* cells using dual-color marked chromosomes.

**Figure supplement 5**. Recombination in *CUP-CLB3* cells.

**Figure supplement 6**. Localization of Rad51 in *CUP-CLB3* cells.

**Figure supplement 7**. Localization of Zip1 in *CUP-CLB3* cells.

**Figure supplement 8**. Preventing homologous recombination does not affect the phenotypes caused by premature *CLB3* expression.

pachytene using the *NDT80* block-release system. In this system, expression of *NDT80* is controlled by the *GAL1-10* promoter, which is regulated by an estrogen-inducible Gal4-ER fusion (***Benjamin et al., 2003***; ***Carlile and Amon, 2008***). Cells were induced to sporulate and after 135 min, copper was added to induce cyclin expression. 4 hr 30 min after sporulation induction, estrogen was added to allow cells to synchronously proceed through the meiotic divisions. In wild-type, *CUP-CLB4* and *CUP-CLB5* cells, sister chromatids cosegregated in the first division, resulting in binucleate cells with a GFP dot in one of the two nuclei. In contrast, 39% of *CUP-CLB1* and 41% of *CUP-CLB3* cells segregated sister chromatids in the first division, as judged by the presence of binucleate cells with a GFP dot in each nucleus (***Figure 2D***). We observed a similar result for chromosome III and cells in which one copy of chromosome V was marked with a GFP dot and the other copy with an RFP dot (***Figure 2— figure supplements 3 and 4***).

To confirm that sister chromatids indeed split during meiosis I in cells expressing *CLB3* during prophase I, we examined when sister chromatid separation occurred with respect to Securin (Pds1 in budding yeast) degradation in *CUP-CLB3* cells. In wild-type cells harboring heterozygous CENV-GFP dots, Pds1 degradation was immediately followed by movement of the single GFP dot to one side of the cell, indicating that homologous chromosomes had segregated. Subsequently, these cells underwent meiosis II and sister chromatids segregated (median = 86 min after Pds1 degradation; ***Figure 2E***). In contrast, *CUP-CLB3* cells segregated sister chromatids immediately after Pds1 degradation (median = 7 min after Pds1 degradation; ***Figure 2E***). These results demonstrate that *CUP-CLB3* cells

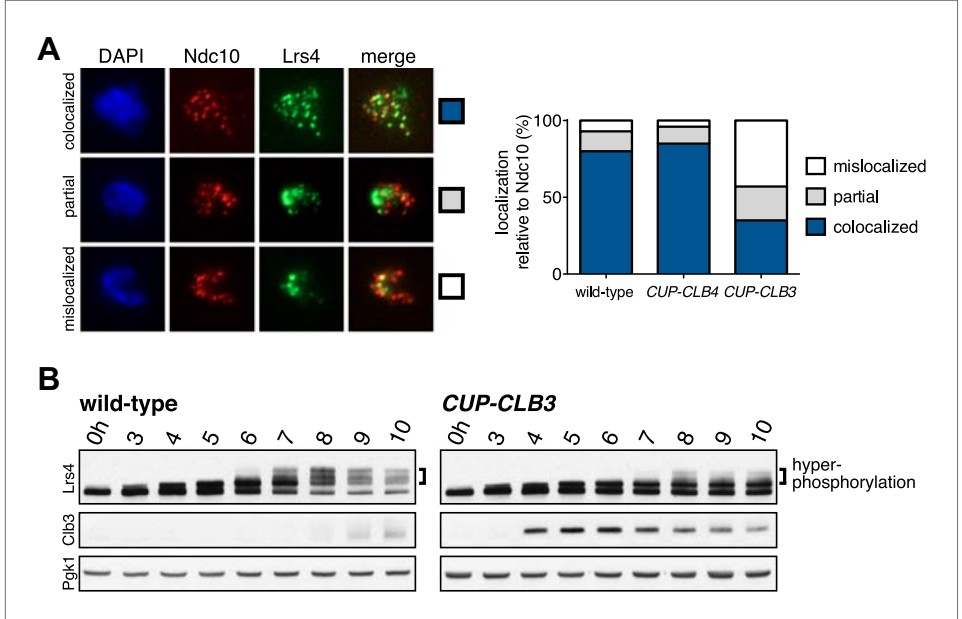

**Figure 3**. Premature *CLB3* expression disrupts monopolin function. (**A**) Lrs4-13myc (green) localization relative to Ndc10-6HA (red) was determined in spread nuclei from wild-type (A9217), *CUP-CLB3* (A26278) and *CUP-CLB4* (A29643) harboring a Cdc20 depletion allele (*cdc20-mn*) were induced to undergo sporulation and arrested in metaphase I due to depletion of Cdc20. CuSO$_4$ was added at 3 hr after induction of sporulation (n > 40). The fraction of spread nuclei that display colocalized, partial or mislocalized Lrs4 with respect to Ndc10 was compared to wild-type using a chi-square test (df 2): *CUP-CLB4*, $\chi^2$ = 1.136, p=0.5666; *CUP-CLB3*, $\chi^2$ = 45.84, p<0.0001. (**B**) Western blots for Lrs4-13myc, Clb3 and Pgk1 from wild-type (A9217) and *CUP-CLB3* (A26278) cells. Cells were sporulated as described in (**A**).

The following figure supplements are available for figure 3.

**Figure supplement 1**. Monopolin association with kinetochores is disrupted in *CUP-CLB3* but not in *CUP-CLB4* cells.

**Figure supplement 2**. Premature Clb3 expression does not interfere with Mam1 expression.

**Figure supplement 3**. Lrs4 phosphorylation is not disrupted in *CUP-CLB4* cells.

segregate sister chromatids during the first meiotic division. Thus, *CUP-CLB3* cells must be defective in two key aspects of meiosis I chromosome segregation: coorientation of sister kinetochores and maintenance of centromeric cohesion. We note that another essential aspect of meiosis I chromosome segregation, homologous recombination, was not affected by premature *CLB3* expression. We observed no major defects in DSB formation, synaptonemal complex assembly and generation of recombination products, nor did preventing homologous recombination affect the phenotypes caused by premature *CLB3* expression (*Figure 2—figure supplements 5–8*).

## Premature expression of *CLB3* interferes with monopolin localization

The finding that *CUP-CLB1* or *CUP-CLB3* cells segregate sister chromatids during meiosis I indicates that sister kinetochore coorientation is defective. To investigate this further, we examined monopolin localization in cells that segregate sister chromatids in meiosis I (*CUP-CLB3* cells) and cells that do not exhibit chromosome missegregation despite cyclin misexpression (*CUP-CLB4* cells). Colocalization of Lrs4 or Mam1 with the kinetochore component Ndc10 was dramatically reduced in *CUP-CLB3* but not *CUP-CLB4* cells (*Figure 3A* and *Figure 3—figure supplements 1 and 2*). Hyperphosphorylation of Lrs4, which correlates with monopolin function (*Clyne et al., 2003*; *Lee and Amon, 2003*; *Matos et al., 2008*), was also significantly reduced in *CUP-CLB3*, but not in *CUP-CLB4* cells (*Figure 3B* and *Figure 3—figure supplement 3*). These results indicate that premature expression of *CLB3* prevents monopolin association with kinetochores.

## Centromeric cohesin is lost during meiosis I in *CUP-CLB3* cells

Sister chromatids segregate during meiosis I in *CUP-CLB3* cells, indicating that centromeric cohesin either fails to associate with chromosomes or is lost prematurely. To test the first possibility, we examined chromosome association of the cohesin subunit Rec8 and the cohesion maintenance factor Pds5 with chromosomes. Chromatin immunoprecipitation (ChIP) and chromosome spreads revealed that association of both proteins with chromosomes in *CUP-CLB3* cells was indistinguishable from that of wild-type cells during prophase I or metaphase I (*Figure 4A* and *Figure 4—figure supplements 1 and 2*). Thus, loading of cohesion factors onto chromosomes is not affected in *CUP-CLB3* cells.

To test the possibility that *CUP-CLB3* cells fail to maintain centromeric cohesion beyond anaphase I, we first determined the localization of the cohesin subunit Rec8 in cells that had progressed past metaphase I. Rec8 colocalized with the kinetochore component Ndc10 in binucleate wild-type and *CUP-CLB4* cells, demonstrating that centromeric cohesin is protected from removal until the onset of anaphase II. In contrast, Rec8 was not detected around centromeres in a substantial fraction of binucleate *CUP-CLB3* cells (*Figure 4B*). Functional assays confirmed the defect in centromeric cohesion maintenance in *CUP-CLB3* cells. Although *mam1Δ* cells biorient sister chromatids during meiosis I, they delay nuclear division until meiosis II due to the presence of centromeric cohesin (*Toth et al., 2000*; *Rabitsch et al., 2003*). The delay in nuclear division of a *mam1Δ* was partially alleviated by the expression of *CUP-CLB3* (*Figure 4—figure supplement 3*). This partial effect is likely due to not all *CUP-CLB3* cells losing centromeric cohesion prematurely in meiosis I (*Figure 4B*). We conclude that both centromeric and arm cohesin are lost from chromosomes at the onset of anaphase I in *CUP-CLB3* cells.

Next, we investigated the cause of premature centromeric cohesin removal in *CUP-CLB3* cells. Cleavage of cohesin by separase requires the phosphorylation of Rec8 at multiple residues (*Brar et al., 2006*; *Katis et al., 2010*). A recessive allele of *REC8* in which 29 in vivo phosphorylation sites were mutated to alanine (*rec8-29A*) (*Brar et al., 2006*) was not cleaved in *CUP-CLB3* cells, but wild-type Rec8 was (*Figure 4C* and *Figure 4—figure supplement 4*). Furthermore, the *rec8-29A* allele caused a similar metaphase I delay in wild-type and *CUP-CLB3* cells when expressed as the sole source of *REC8* (*Figure 4D* and *Figure 4—figure supplements 5 and 6*). We noticed that the Rec8 cleavage product was detected at lower levels in *CUP-CLB3* cells (*Figure 4C* and *Figure 4—figure supplement 5*). The cause of this reduction is currently unclear, but could indicate that in *CUP-CLB3* cells, cohesin removal also relies on a separase-independent pathway, that is the prophase removal pathway (*Yu and Koshland, 2005*).

Our results demonstrate that Rec8 phosphorylation is required for cohesin removal in *CUP-CLB3* cells and suggest that the defect in centromeric cohesin protection may result from increased phosphorylation of centromeric Rec8. To test this possibility, we used phospho-specific antibodies against two in vivo phosphorylation sites of Rec8 (pS179 and pS521) (*Brar et al., 2006*; *Katis et al., 2010*; M. Attner personal communication, October 2011) and analyzed the relative enrichment of total Rec8 and phospho-Rec8 at CENV or at an arm cohesin binding site by ChIP in metaphase I-arrested cells. The two phospho-specific antibodies immunoprecipitated similar amounts of Rec8 in wild-type and *CUP-CLB3* cells at the arm site (*Figure 4E*), which is consistent with arm cohesin being primed for Separase cleavage. However, the amount of phosphorylated Rec8 was increased at the centromere in *CUP-CLB3* cells compared to wild-type cells, albeit not to the same extent as in cells depleted for Sgo1 (*sgo1-mn*), in which meiosis I centromeric-cohesin protection is completely defective (*Figure 4E*). We conclude that *CUP-CLB3* cells are compromised in preventing centromeric Rec8 phosphorylation during meiosis I.

## Sgo1-PP2A localization is not affected in *CUP-CLB3* cells

Sgo1-PP2A and the meiosis-specific protein Spo13 prevent centromeric Rec8 phosphorylation during meiosis I to protect this cohesin pool from cleavage. All three proteins localize to kinetochores during meiosis I, which is thought to be critical for their cohesin-protective function (*Katis et al., 2004a*, *2004b*; *Kitajima et al., 2004*; *Lee et al., 2004*; *Kitajima et al., 2006*; *Riedel et al., 2006*). Surprisingly, Sgo1, the PP2A regulatory subunit Rts1 and Spo13 localized normally in prophase I- and metaphase I-arrested *CUP-CLB3* cells (*Figure 4F–H* and *Figure 4—figure supplements 7–9*). We noticed a moderate reduction of Sgo1 and Rts1 at centromeres in binucleate *CUP-CLB3* cells (*Figure 4I* and *Figure 4—figure supplement 10*). However, this reduction during anaphase I is most likely a consequence

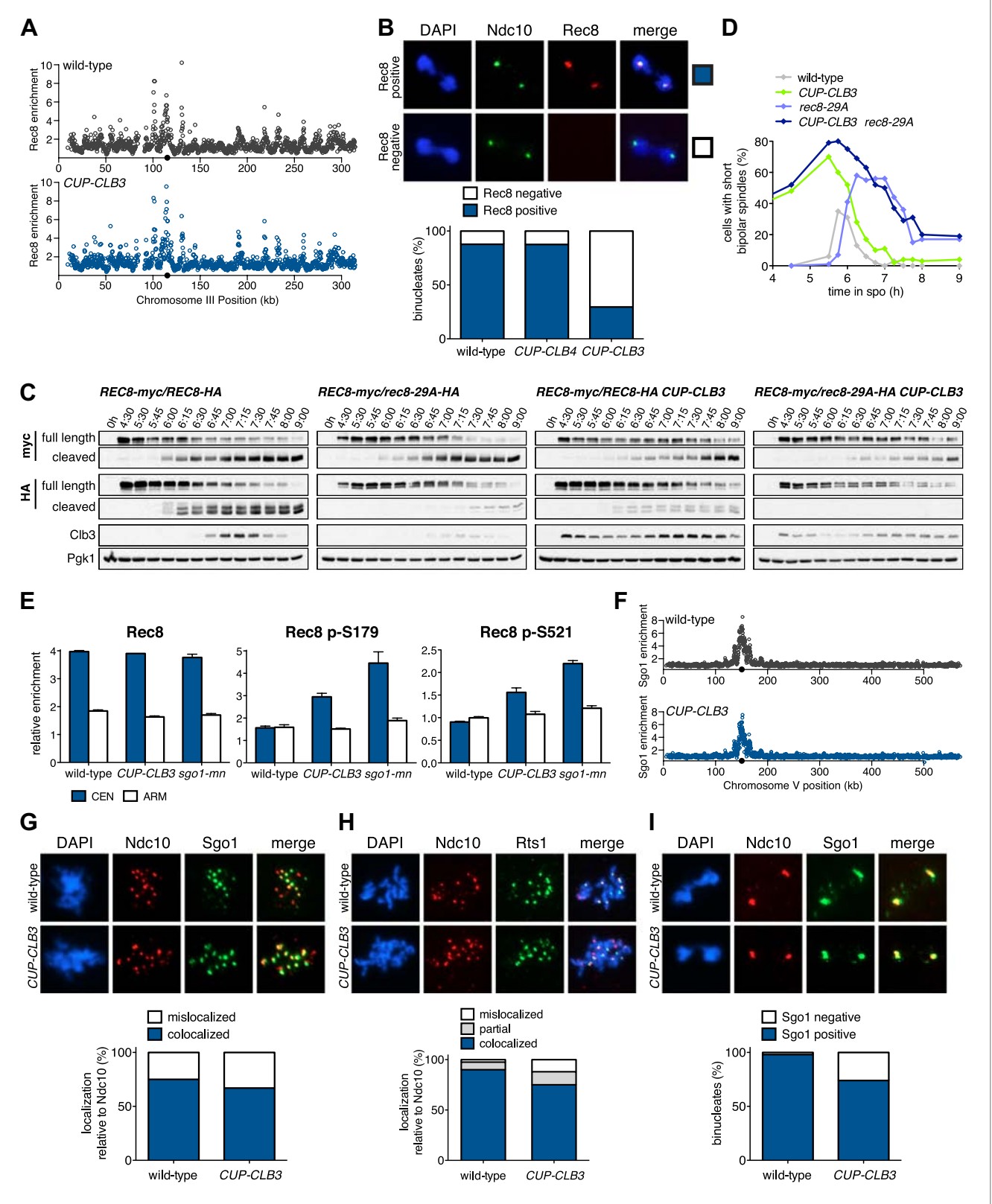

**Figure 4**. *CLB3* misexpression disrupts protection of centromeric cohesin. Cyclin expression was induced after 2 hr 15 min (**C**) and (**D**), 2 hr 30 min (**A**), (**B**), (**E**), (**F**) and (**H**) or 3 hr (**G**) and (**I**) of sporulation. (**A**) Chromosomal association of Rec8-13myc was monitored by ChIP-chip in wild-type (A28716) and *CUP-CLB3* (A28718) during prophase I arrest. Centromere position is identified by a black circle. (**B**) Centromeric Rec8 localization was monitored in
*Figure 4. Continued on next page*

*Figure 4. Continued*

spread nuclei from wild-type (A28684), *CUP-CLB3* (A28685) and *CUP-CLB4* (A28686) cells carrying *REC8-3HA* (red) and *NDC10-13myc* (green) (n > 40). The fraction of spread nuclei that were Rec8 positive or negative was compared to wild-type using a chi-square test (df 1): *CUP-CLB4*, $\chi^2$ = 0.001323, p=0.9710; *CUP-CLB3*, $\chi^2$ = 32.79, p<0.0001. (**C**) Rec8 cleavage monitored by Western blot after release from an *NDT80* block (4 hr 30 min) in wild-type and *CUP-CLB3* carrying both a myc-tagged *REC8* allele as well as either HA-tagged *REC8* or *rec8-29A* allele (left to right: A29957, A29959, A29961, A29963). (**D**) Percentage of cells with short bipolar spindles was determined at indicated times in wild-type (A22804), *CUP-CLB3* (A29965), *rec8-29A* (A22803) and *CUP-CLB3 rec8-29A* (A29967) after release from an *NDT80* block (4 hr 30 min) (n = 100 per time point). (**E**) ChIP analysis for total Rec8, p-S179 Rec8 or p-S521 Rec8 from metaphase I-arrested (*cdc20-mn*) wild-type (A28681), *CUP-CLB3* (A28682) and Sgo1-depleted (*sgo1-mn*; A29994) cells. Relative occupancy at a chromosome arm site (c194) or at a centromeric site (CENV) was determined relative to a low binding region (c281). Error bars represent range (n = 2). (**F**) Chromosomal association of Sgo1-3V5 was monitored by ChIP-chip in wild-type (A29795) and *CUP-CLB3* (A29799) cells during prophase I-arrest. Centromere position is identified by a black circle. (**G**), (**H**) Localization of Sgo1-9myc (G, green) or Rts1-13myc (H, green) relative to Ndc10-6HA (red) determined by nuclear spreads in (**G**) wild-type (A22868) and *CUP-CLB3* (A22870) or (**H**) wild-type (A28329) and *CUP-CLB3* (A28330) during prophase I (n > 40). For (**G**), the fraction of spread nuclei that display colocalized or mislocalized Sgo1 relative to Ndc10 was compared between wild-type and *CUP-CLB3* using a chi-square test (df 1) $\chi^2$ = 1.554, p=0.2125. For (**H**), the fraction of spread nuclei that display colocalized, partial or mislocalized Rts1 relative to Ndc10 was compared between wild-type and *CUP-CLB3* using a chi-square test (df 2) $\chi^2$ = 3.712, p=0.1563. (**I**) Localization of Sgo1-9myc (green) in binucleates relative to Ndc10-6HA (red) determined by nuclear spreads from wild-type (A22868) and *CUP-CLB3* (A22870) (n > 40). The fraction of spread nuclei that were Sgo1 positive or negative was compared between wild-type and *CUP-CLB3* using a chi-square test (df 1) $\chi^2$ = 23.92, p<0.0001.

The following figure supplements are available for figure 4.

**Figure supplement 1**. Chromosomal association of Rec8 in *CUP-CLB3* cells.

**Figure supplement 2**. Chromosomal association of Pds5 in *CUP-CLB3* cells.

**Figure supplement 3**. *CUP-CLB3* cells partially bypass the nuclear division delay of *mam1Δ* cells.

**Figure supplement 4**. Meiotic progression of the cells analyzed for Rec8 cleavage in ***Figure 4C***.

**Figure supplement 5**. Analysis of Rec8 cleavage in cells used for ***Figure 4D***.

**Figure supplement 6**. Meiotic progression of the cells analyzed for Rec8 cleavage in ***Figure 4D***.

**Figure supplement 7**. Chromosomal association of Sgo1 in *CUP-CLB3* cells.

**Figure supplement 8**. Localization of Rts1 in *CUP-CLB3* cells.

**Figure supplement 9**. Chromosomal association of Spo13 in *CUP-CLB3* cells.

**Figure supplement 10**. Rts1 localization in binucleate *CUP-CLB3* cells.

**Figure supplement 11**. Analysis of Rts1 localization in Rec8 phosphomimetic mutants.

rather than a cause of premature loss of centromeric cohesin. In cells expressing a phosphomimetic version of Rec8 (*rec8-4D*) that cannot be retained at centromeres beyond meiosis I, Rts1 localization is also reduced in anaphase I (***Figure 4—figure supplement 11***). It is thus unlikely that the reduction of Sgo1 and Rts1 at centromeres during anaphase I contributes to the premature loss of centromeric cohesin. These findings, together with our observation that centromeric Rec8 phosphorylation is increased in *CUP-CLB3* cells, indicate that Sgo1-PP2A function, but not localization, is impaired in *CUP-CLB3* cells.

## Modulating microtubule–kinetochore interactions affects monopolin-induced sister chromatid cosegregation during mitosis

How does premature expression of *CLB3* interfere with establishment of the meiosis I chromosome segregation pattern? The comparison of the effects caused by *CLB3* and *CLB4* misexpression provided insight into this question. Both cyclins induce spindle formation in prophase I. However, chromosomes are able to attach to this spindle and experience pulling forces only in *CUP-CLB3* cells. Thus,

the ability to form tension-generating attachments (i.e. *CUP-CLB1* or *CUP-CLB3* cells) correlates with defects in meiosis I chromosome morphogenesis and segregation. This correlation suggests that premature microtubule–kinetochore engagement during premeiotic S phase/early prophase I is the underlying cause of chromosome missegregation in *CUP-CLB3* cells and predicts that tension generating microtubule–kinetochore attachments should inhibit meiosis I chromosome morphogenesis. Conversely, preventing them should enable building a proper meiosis I chromosome architecture.

We tested the first prediction using a previously described method in which monopolin-dependent sister kinetochore coorientation is induced during mitosis (*Monje-Casas et al., 2007*). Overexpression of *MAM1* and *CDC5* upon a pheromone-induced G1 arrest is sufficient to induce cosegregation of sister chromatids in mitotic anaphase (*Monje-Casas et al., 2007*, *Figure 5A*). However, when cells are allowed to form microtubule–kinetochore attachments prior to *CDC5* and *MAM1* expression, cosegregation of sister chromatids is prevented. We reversibly arrested cells in metaphase using a methionine repressible *CDC20* allele (*MET-CDC20*) and induced *MAM1* and *CDC5* expression after cells had arrested in metaphase and had formed microtubule–kinetochore interactions. Under these conditions, *MAM1* and *CDC5* expression did not induce sister chromatid cosegregation when cells were released into anaphase (*Figure 5A*). Importantly, disrupting microtubule–kinetochore interactions by depolymerizing microtubules with nocodazole during the metaphase arrest resulted in robust cosegregation of sister chromatids in anaphase (48% cosegregation, *Figure 5A*). These results show that microtubule–kinetochore interactions modulate the ability of monopolin to induce sister chromatid cosegregation.

## Transient disruption of microtubule–kinetochore interactions restores meiosis I chromosome segregation in *CUP-CLB3* cells

If the defects in sister kinetochore coorientation and centromeric cohesin maintenance of *CUP-CLB3* cells are caused by premature microtubule–kinetochore interactions, proper meiosis I chromosome morphogenesis should be restored by transiently disrupting microtubule–kinetochore interactions. To test this, we used a temperature sensitive allele of *NDC80* (*ndc80-1*), which encodes a component of the outer kinetochore. *ndc80-1* cells grow and sporulate normally at 25°C, but fail to undergo any nuclear divisions at temperatures above 34°C (*Figure 5—figure supplement 1*).

We first asked whether disrupting microtubule–kinetochore interactions suppresses the kinetochore localization defect of monopolin in *CUP-CLB3* cells. Using the *NDT80* block-release system, we induced cells to sporulate at 25°C. After 165 min, we induced cyclin expression and concurrently transferred cells to 34°C to inactivate the *ndc80-1* allele. Cells were then incubated for an additional 135 min to arrest them in the *NDT80*-depletion block. We then transferred cells to the permissive temperature and released them from the *NDT80* block into a metaphase I-arrest by depleting *CDC20* (*cdc20-mn*) (*Figure 5B*). Under these conditions, wild-type and *ndc80-1* cells arrested in metaphase I with the monopolin subunit Lrs4 localized to kinetochores, while *CUP-CLB3* cells showed a defect in Lrs4 localization (*Figure 5C*). Remarkably, *CUP-CLB3 ndc80-1* cells showed near wild-type levels of Lrs4 association with kinetochores (*Figure 5C*). Transient inactivation of *Ndc80* also restored Lrs4 phosphorylation in *CUP-CLB3* cells (*Figure 5D*). Our results demonstrate that premature microtubule–kinetochore interactions prevent sister kinetochore coorientation by disrupting proper localization of the monopolin complex. The finding that transient disruption of microtubule–kinetochore interactions also suppresses the Lrs4 phosphorylation defect of *CUP-CLB3* cells, furthermore suggests that Lrs4 hyperphosphorylation occurs not at the time of nucleolar release, but once Lrs4 localizes to kinetochores.

We next asked whether transient inactivation of microtubule–kinetochore interactions also suppresses the premature loss of centromeric cohesin observed in *CUP-CLB3* cells. We used a similar protocol to the one described above, except cells were not arrested in metaphase I following release from the *NDT80* block, but were allowed to proceed into anaphase I to examine Rec8 localization. Remarkably, disrupting microtubule–kinetochore interactions at the time of Clb3 expression caused a considerable increase in the percentage of *CUP-CLB3* cells that retained Rec8 around centromeres during anaphase I (*Figure 5E*).

Finally, restoring centromeric cohesin protection and sister kinetochore coorientation to *CUP-CLB3* cells by transient inactivation of *Ndc80* restored homolog segregation during meiosis I (*Figure 5F* and *Figure 5—figure supplements 2–4*). Similar results were obtained with a temperature sensitive allele of the gene encoding the outer kinetochore component Dam1 (*dam1-1*) or by disrupting microtubule–kinetochore interactions by benomyl treatment (*Figure 5—figure supplements 5 and 6*). We

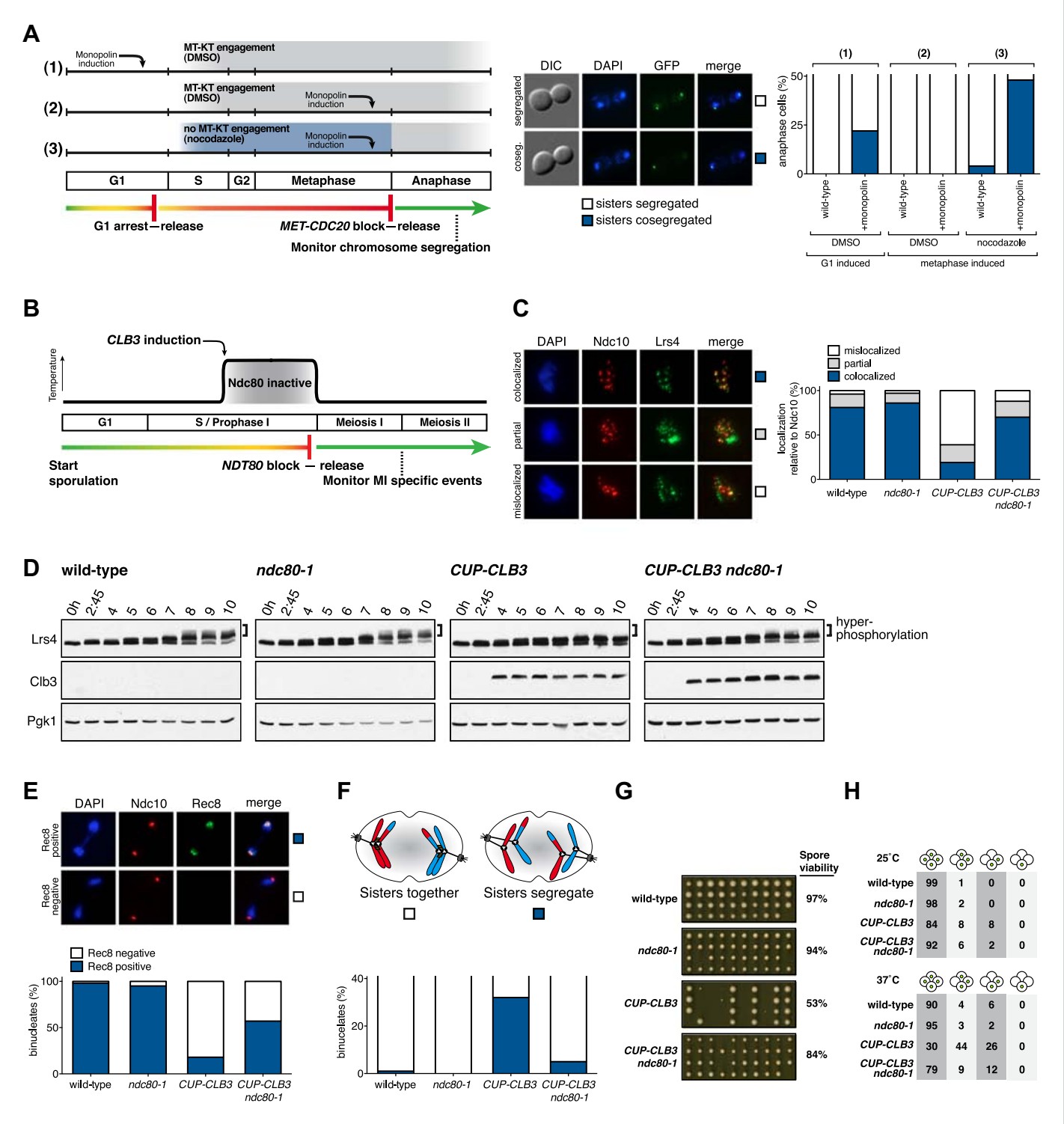

**Figure 5**. Transient disruption of microtubule–kinetochore interactions suppresses the chromosome segregation defects in *CUP-CLB3* cells. (**A**) Wild-type (A10684) and *GAL-CDC5 GAL-MAM1* (A26546) cells, carrying a *MET-CDC20* allele and CENIV-GFP dots, were monitored for chromosome segregation in anaphase (see 'Materials and methods' for details). MT = microtubule, KT = kinetochore, (n = 100). The fraction of anaphase cells that segregate or cosegregate sister chromatids was compared between *GAL-CDC5 GAL-MAM1* condition (2) and *GAL-CDC5 GAL-MAM1* condition (3) using a chi-square test (df 1) χ² = 59.71, p<0.0001. (**B**) Schematic description of the experimental regime used for (**C**) through (**H**) see 'Materials and methods' for details. (**C**) Localization of Lrs4-13myc (green) in mononucleates relative to Ndc10-6HA (red) determined by nuclear

*Figure 5. Continued on next page*

*Figure 5. Continued*

spreads (n > 40) and (**D**) phosphorylation of Lrs4-13myc determined by gel mobility shift in wild-type (A29612), *ndc80-1* (A29614), *CUP-CLB3* (A29616) and *CUP-CLB3 ndc80-1* (A29618). For (**C**), using a chi-square test (df 2) the fraction of spread nuclei that display colocalized, partial or mislocalized Lrs4 with respect to Ndc10 was compared between wild-type and *ndc80-1* $\chi^2$ = 0.9668, p=0.6167 and between *CUP-CLB3* and *CUP-CLB3 ndc80-1* $\chi^2$ = 56.34, p<0.0001. (**E**) Localization of Rec8-13myc (green) in binucleates relative to Ndc10-6HA (red) determined by nuclear spreads in wild-type (A28716), *ndc80-1* (A28720), *CUP-CLB3* (A28718) and *CUP-CLB3 ndc80-1* (A28722) (n > 40). Using a chi-square test (df 1) the fraction of spread nuclei that were Rec8 positive or negative was compared between wild-type and *ndc80-1* $\chi^2$ = 1.185, p=0.2764 and between *CUP-CLB3* and *CUP-CLB3 ndc80-1* $\chi^2$ = 23.96, p<0.0001. (**F**) Segregation of sister chromatids using heterozygous CENV-GFP dots quantified in binucleates (n = 100) and (**G**) spore viability from wild-type (A22678), *ndc80-1* (A28621), *CUP-CLB3* (A22702) and *CUP-CLB3 ndc80-1* (A28623) (n = 40 tetrads for wild-type and *ndc80-1*, n > 60 tetrads for *CUP-CLB3* and *CUP-CLB3 ndc80-1*) (nonpermissive temperature >36°C). Using a chi-square test (df 1) the fraction of binucleates with a reductional or equational division was compared between *CUP-CLB3* and *CUP-CLB3 ndc80-1* $\chi^2$ = 24.18, p<0.0001. (**G**) Segregation of chromosome V using homozygous CENV-GFP dots quantified in tetranucleates from wild-type (A22688), *ndc80-1* (A28625), *CUP-CLB3* (A22708) and *CUP-CLB3 ndc80-1* (A28627). Top panel: cells kept at 25°C for the duration of the experiment. Bottom panel: Cells treated as in (**B**) but monitored after meiosis II (n = 100).

The following figure supplements are available for figure 5.

**Figure supplement 1**. Sporulation efficiency of *ndc80-1* mutants.

**Figure supplement 2**. Sister kinetochore coorientation in *ndc80-1* cells under a continuous inactivation regime at 34°C during a metaphase I arrest.

**Figure supplement 3**. Sister kinetochore coorientation in *ndc80-1* cells after transient inactivation regime at 34°C during a metaphase I arrest.

**Figure supplement 4**. Meiosis I chromosome segregation in *ndc80-1* cells after a transient inactivation regime at 34°C.

**Figure supplement 5**. Transient disruption of microtubule–kinetochore interactions using *dam1-1* allele restores meiosis I chromosome segregation in *CUP-CLB3* cells.

**Figure supplement 6**. Transient disruption of microtubule–kinetochore interactions by benomyl treatment restores meiosis I chromosome segregation in *CUP-CLB3* cells.

**Figure supplement 7**. Transient disruption of microtubule–kinetochore interactions during S phase/prophase I suppresses *CUP-CLB3*-induced meiosis I sister chromatid segregation in a spindle assembly checkpoint independent manner.

**Figure supplement 8**. Transient disruption of microtubule–kinetochore interactions during S phase/prophase I restores meiotic chromosome segregation in *CUP-CLB3* cells in a spindle assembly checkpoint independent manner.

**Figure supplement 9**. Transient *ndc80-1* inactivation does not alter in vitro Cdk1 activity.

further observed a striking improvement in overall chromosome segregation and spore viability in *CUP-CLB3 ndc80-1* compared to *CUP-CLB3* cells (***Figure 5G,H***). The suppression of chromosome missegregation in *CUP-CLB3 ndc80-1* cells did not depend on the spindle assembly checkpoint, because deletion of *MAD3* had no discernable effect on the extent of *ndc80-1* mediated suppression (***Figure 5—figure supplements 7 and 8***), nor was it due to the *ndc80-1* allele lowering Clb3-CDK activity (***Figure 5—figure supplement 9***). In summary, our results demonstrate that the defects associated with *CUP-CLB3* cells are due to premature microtubule–kinetochore interactions. Our results further suggest that preventing microtubule–kinetochore interactions during premeiotic S phase and prophase I is necessary to establish a meiosis I-specific chromosome architecture.

## The outer kinetochore is disassembled during premeiotic S phase and prophase I

Our results demonstrate that preventing premature interactions of microtubules with kinetochores is essential for establishing a meiosis I chromosome architecture. This occurs, at least in part, by restricting Clb-CDK activity during premeiotic S phase and prophase I. Are additional mechanisms in place to prevent premature microtubule–kinetochore interactions? Insight into this question came from the variability in *CUP-CLB3*-associated phenotypes.

We initially noticed that the timing of *CLB3* induction had an impact on the extent of sister chromatid segregation in meiosis I, especially in experiments that employed the *NDT80* block-release system. To investigate this further, we expressed *CLB3* at different times after induction of sporulation. We observed that the extent of meiosis I sister chromatid segregation declined as *CLB3* was expressed later during the *NDT80* block (*Figure 6A*). One possibility is that *CLB3*-induced sister chromatid segregation depends on additional factors that become limiting. Kinetochore components are good candidates for such additional factors, because previous studies in fission yeast demonstrated that a subset of outer kinetochore components dissociates from the kinetochore during prophase I (*Asakawa et al., 2005*).

Using a high-resolution ribosome profiling dataset (*Brar et al., 2012*), we examined the timing of synthesis of all kinetochore components during meiotic progression by cluster analysis. This analysis revealed two major expression classes, one included kinetochore components that peak in expression prior to or during prophase I (early class), and the other contained components that instead show peak expression during the meiotic divisions (late class). The early class was enriched for inner kinetochore components (16 of 23), while the late class included primarily outer kinetochore components (13 of 18) (*Figure 6B*, *Figure 6—figure supplements 1–10*). The inner kinetochore binds to the centromere and generates a platform for the assembly of the outer kinetochore, which mediates microtubule attachments (*Tanaka, 2010*). The temporal difference in expression suggests that the inner kinetochore is assembled prior to the meiotic divisions, while the outer kinetochore is constructed only as cells enter the meiotic divisions.

Among the outer kinetochore components that displayed peak synthesis during the divisions, *NDC80* and a subunit of the DASH complex, *HSK3,* displayed the most differential expression prior to meiosis I and during meiosis I, with a 9.02 and 2.64-fold induction, respectively (*Figure 6B–D*). This decline in Ndc80 expression is consistent with a previous study in fission yeast, showing that Ndc80 becomes undetectable at kinetochores during prophase I (*Asakawa et al., 2005*). Analysis of Ndc80 protein levels provided an explanation for why cells upregulate the synthesis of outer kinetochore components during entry into meiosis I. Ndc80 levels declined during premeiotic S phase and became undetectable during late prophase I (*Figure 6E*). Importantly, the ability of *CUP-CLB3* to induce sister-chromatid segregation during meiosis I tightly correlated with Ndc80 protein levels; as Ndc80 protein declines, so does *CLB3*-induced meiosis I sister chromatid segregation (compare *Figure 6A,E*).

Hsk3 protein levels were also low until meiosis I entry (*Figure 6F*), but not all outer kinetochore components exhibited this decline in protein levels. For example, Ask1, a subunit of the DASH complex, was present throughout prophase I and levels of another DASH complex component, Dam1, increased during prophase I (*Figure 6—figure supplement 11*). Our findings indicate that the assembly of the outer kinetochore is restricted prior to *NDT80* expression and pachytene exit due to low levels of a subset of outer kinetochore components.

## Expression of *NDC80* and *HSK3* during premeiotic S phase/prophase I enhances *CLB3*-induced meiosis I sister chromatid segregation

To determine whether reduced expression of the outer kinetochore components Ndc80 and Hsk3 contributes to preventing premature microtubule–kinetochore engagement, we examined the consequences of expressing the two genes from the *CUP1* promoter (*Figure 7*). We first assessed whether expression of the two proteins allows for the recruitment of the DASH complex to kinetochores, which occurs via delivery through microtubules and can thus be used as a means of assessing end-on attachment of kinetochores (*Cheeseman et al., 2001*; *Tanaka, 2010*). Cells were induced to sporulate and after 4 hr, a time when Ndc80 levels are normally diminished, we induced the expression of *CLB3*, *NDC80* and/or *HSK3*. Whereas expression of either gene alone caused only a few cells to recruit Ask1 to kinetochores, cells simultaneously expressing *NDC80*, *HSK3* and *CLB3* during prophase I showed colocalization between Ask1 and the inner kinetochore component Ndc10, to an equal or greater extent than what was observed in metaphase I-arrested wild-type cells (*Figure 8A,B*). The difference in Ask1 localization was not due to a difference in *ASK1* expression (*Figure 8C*). In addition, induction of *CLB3* under the conditions mentioned above gave rise to bipolar spindles that appeared fragile with a weakened midzone. In contrast, consistent with stable microtubule–kinetochore interactions, coexpression of *CLB3*, *HSK3* and *NDC80* resulted in the formation of robust bipolar spindles (*Figure 8D,E*). Importantly, the expression of *NDC80* and/or *HSK3* during an *NDT80* block caused a considerable

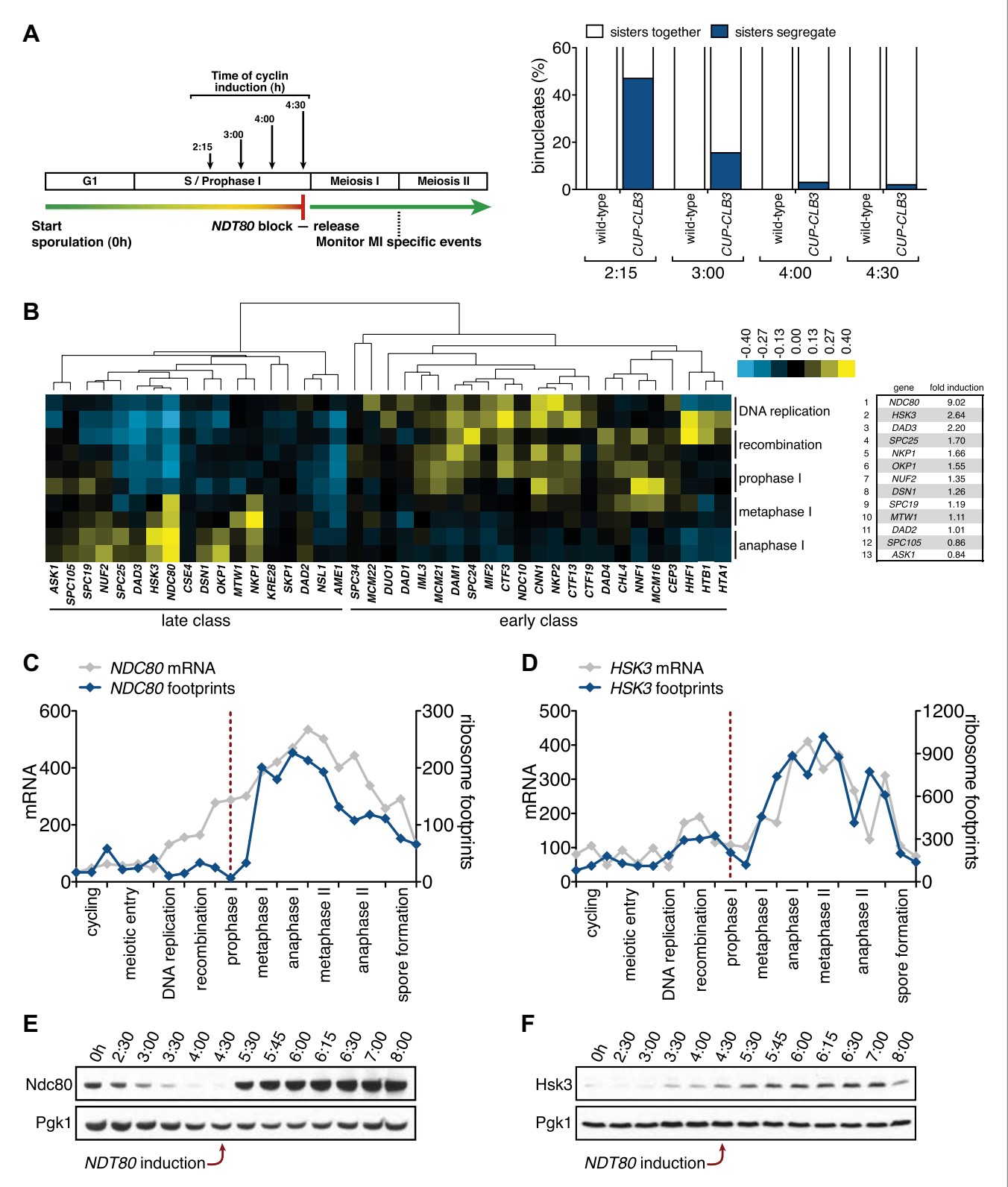

**Figure 6**. Meiosis I sister chromatid segregation correlates with presence of outer kinetochore components. (**A**) Schematic description of the experimental regime and segregation of sister chromatids using heterozygous CENV-GFP dots quantified in binucleates from wild-type (A22678) and *CUP-CLB3* (A29406) after cyclin induction at 2 hr 15 min, 3 hr, 4 hr and 4 hr 30 min post transfer to sporulation medium. Cells released from *NDT80* block at 4 hr 30 min

*Figure 6. Continued on next page*

*Figure 6. Continued*

(n = 100). Using a chi-square test (df 1), the fraction of binucleates that display a reductional or equational division was compared between wild-type and *CUP-CLB3* for each induction time point: (2:15), $\chi^2$ = 58.00, p<0.0001; (3:00), $\chi^2$ = 14.46, p=0.0001; (4:00), $\chi^2$ = 1.020, p=0.3124; (4:30), $\chi^2$ = 0.3384, p=0.5607. (**B**) Cluster analysis of kinetochore components from the indicated time points. Further details are in the 'Materials and methods' and in *Brar et al. (2012)*. Inner kinetochore = Cse4 nucleosomes, Cbf3, Ctf19 complexes and Mif2. Outer kinetochore = Spc105, Mis12, Ndc80 and DASH complexes. Fold induction is calculated by dividing the average expression from metaphase I—anaphase I by the average expression from DNA replication-prophase I. (**C**) Ordered plot for mRNA-seq and ribosome footprinting data for *NDC80* and (**D**) *HSK3* at the indicated stages. Dotted line indicates time of release from *NDT80* block. Further details are in the 'Materials and methods' and in *Brar et al. (2012)*. (**E**) Western blot for Ndc80-3V5 and Pgk1 from A30340 cells and (**F**) Hsk3-3V5 and Pgk1 from A31861 cells. Cells induced to sporulate and released from *NDT80* block at 4 hr 30 min.

The following figure supplements are available for figure 6.

**Figure supplement 1**. Schematic representation of the kinetochore–microtubule interface.

**Figure supplement 2**. Meiotic cluster analysis of kinetochore components.

**Figure supplement 3**. Meiotic expression of DASH complex subunits.

**Figure supplement 4**. Meiotic expression of Ndc80 complex subunits.

**Figure supplement 5**. Meiotic expression of Mif2.

**Figure supplement 6**. Meiotic expression of KNL-1 complex subunits.

**Figure supplement 7**. Meiotic expression of Mis12 complex subunits.

**Figure supplement 8**. Meiotic expression of Ctf19 complex subunits.

**Figure supplement 9**. Meiotic expression of Cbf3 complex subunits.

**Figure supplement 10**. Meiotic expression of Histone subunits.

**Figure supplement 11**. Meiotic expression of Dam1 and Ask1.

increase in meiosis I sister chromatid segregation in *CUP-CLB3* cells (*Figure 8F*). Furthermore, under conditions in which *CLB3* expression alone failed to induce meiosis I sister chromatid segregation, expression of *CLB3* together with *NDC80* and *HSK3* caused a substantial increase in meiosis I sister chromatid segregation (*Figure 8G*). This occurred even when cells were maintained in a prolonged *NDT80* block prior to expression of *CLB3*, *NDC80* and *HSK3* (*Figure 9*), ruling out the possibility that the expression of *NDT80* targets, such as *CDC5*, early during sporulation contributes to sister chromatid segregation during meiosis I. We conclude that limiting outer kinetochore assembly is an additional mechanism to prevent microtubule–kinetochore interactions during premeiotic S phase and prophase I.

## Discussion

The specialized chromosome segregation pattern in meiosis likely evolved through modifications of the mitotic cell division program. We find that preventing microtubule–kinetochore interactions during premeiotic S phase and prophase I is essential for transforming mitosis into meiosis I. Meiosis I chromosome morphogenesis, including the assembly of cohesin protective structures around centromeres and sister kinetochore coorientation, occurs during prophase I. We propose that when microtubules interact with kinetochores prior to completion of this remodeling process, they establish a default attachment, biorientation, which is incompatible with establishing sister kinetochore coorientation and a cohesin protective domain around centromeres (*Figure 10*). Our findings reveal a novel regulatory event that is essential for accurate meiosis I chromosome segregation and demonstrate that temporal restriction of microtubule–kinetochore interactions is instrumental in transforming mitosis into meiosis.

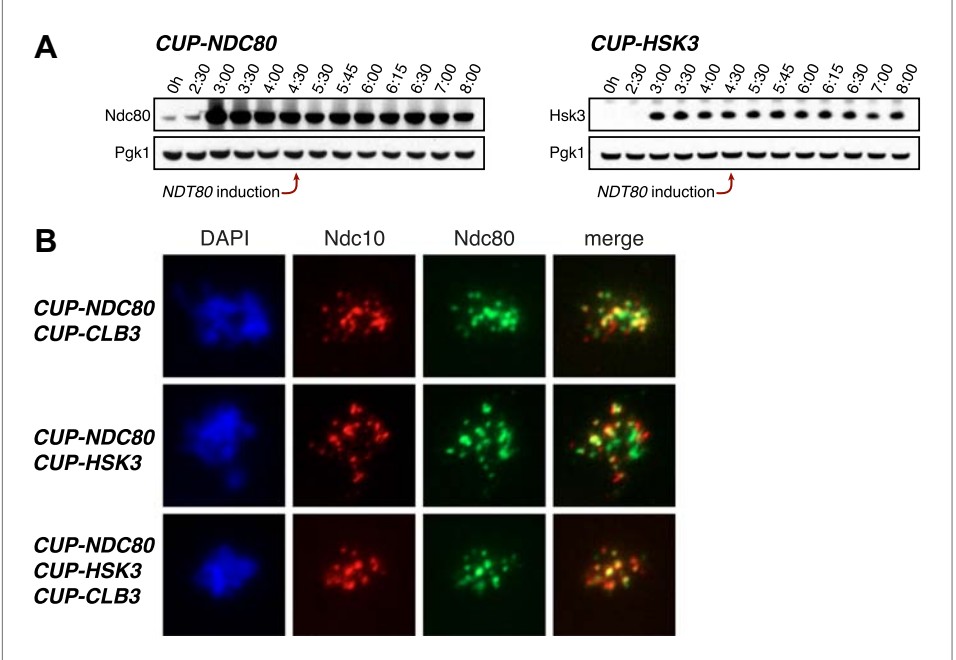

**Figure 7**. Characterization of *NDC80* and *HSK3* overexpression. (**A**) *CUP-NDC80-3V5* (A30342) and *CUP-HSK3-3HA* (A32060) cells also carrying the *GAL4-ER* and *GAL-NDT80* fusions were induced to sporulate. After 2 hr 30 min CuSO$_4$ (50 µM) was added and cells were subsequently released from *NDT80* block 4 hr 30 min after transfer into sporulation medium. The levels of Ndc80-3V5, Hsk3-3HA and Pgk1 were monitored by Western blot. (**B**) *CUP-NDC80-3V5 CUP-CLB3* (A31949), *CUP-NDC80-3V5 CUP-HSK3* (A31951) and *CUP-NDC80-3V5 CUP-HSK3 CUP-CLB3* (A31953) cells were induced to sporulate. 4 hr after transfer into sporulation medium CuSO$_4$ (50 µM) was added, and localization of Ndc80-3V5 (green) relative to Ndc10-6HA (red) was determined by nuclear spreads 5 hr after transfer into sporulation medium.

## The effects of premature microtubule–kinetochore engagement on meiosis I chromosome morphogenesis

Transcriptional and translational controls restrict *CLB3* expression to meiosis II (*Carlile and Amon, 2008*). Eliminating both, by placing the gene under the control of the *GAL1-10* promoter or the *CUP1* promoter has dramatic effects on meiosis I chromosome segregation. *CLB3* expression from the *GAL1-10* promoter, which leads to Clb3 levels comparable to those seen for wild-type cells in meiosis II, causes a significant suppression of the meiosis I chromosome segregation pattern. This defect is not further enhanced by overexpression of the protein (by expression from the *CUP1* promoter), which further indicates that this phenotype does not emanate from expressing exceedingly high levels of the cyclin, but is a consequence of premature expression.

The consequences of premature *CLB3* expression are dramatic. It leads to premature microtubule–kinetochore interactions and prevents coorientation factors from associating with kinetochores. The observation that the transient disruption of microtubule–kinetochore interactions, either by inactivating the outer kinetochore or by microtubule depolymerization, allowed coorientation factors to associate with kinetochores, despite *CLB3* misexpression, led us to conclude that it is premature microtubule–kinetochore interactions that interfere with the establishment of sister kinetochore coorientation during meiosis I. It is currently unclear how preexisting microtubule–kinetochore interactions prevent monopolin association with kinetochores. Precocious attachment of microtubules to kinetochores could occlude the monopolin complex from binding to kinetochores. Alternatively, tension between sister kinetochores generated from stable microtubule–kinetochore interactions could induce a conformational change at the kinetochore and/or pericentric chromatin such that coorientation factors can no longer associate with the kinetochore.

In addition to preventing sister kinetochore coorientation, premature expression of *CLB3* interferes with protecting centromeric cohesin from removal during meiosis I. The same logic as outlined for

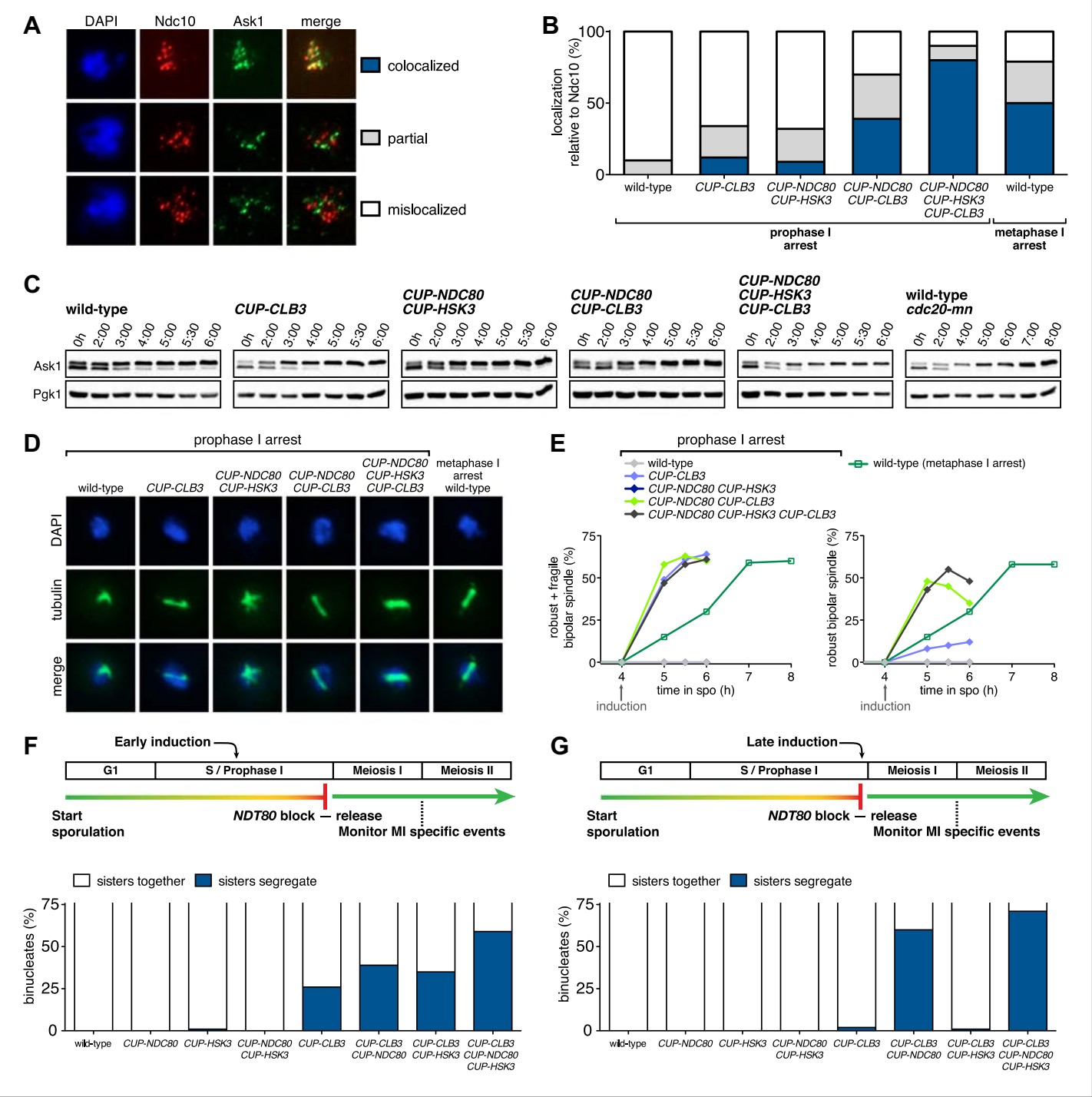

**Figure 8**. Expression of *NDC80* and *HSK3* in prophase I enhances Clb3-CDK-induced meiosis I sister chromatid segregation. For (**A**)–(**E**), wild-type (A31945), *CUP-CLB3* (A31947), *CUP-NDC80 CUP-HSK3* (A31951), *CUP-NDC80 CUP-CLB3* (A31949), *CUP-NDC80 CUP-HSK3 CUP-CLB3* (A31953) and *cdc20-mn* (A31955) cells were induced to sporulate and CuSO₄ (50 μM) was added at 4 hr after sporulation induction. (**A**) Representative images and (**B**) quantification of Ask1-13myc (green) in mononucleates relative to Ndc10-6HA (red) determined by nuclear spreads prepared after 1 hr of CuSO₄ induction (n > 40 except for A31955 [n = 28]). For (**B**), using a chi-square test (df 2) the fraction of spread nuclei that display colocalized, partial or mislocalized Ask1 with respect to Ndc10 was compared between *CUP-CLB3* and *CUP-CLB3 CUP-NDC80 CUP-HSK3* $\chi^2$ = 51.49, p<0.0001. (**C**) Western blots of Ask1-13myc and Pgk1. (**D**) Bipolar spindle morphology and (**E**) left panel, total (robust + fragile) bipolar spindle formation, and right panel, robust bipolar spindle formation determined at the indicated time points (see 'Materials and methods' for further description) (n = 100 per time point). Note: *CUP-NDC80 CUP-HSK3* (dark blue) data points occluded by wild-type (grey) data points. (**F**), (**G**) Segregation of sister chromatids using

*Figure 8. Continued on next page*

*Figure 8. Continued*

heterozygous CENV-GFP dots quantified in binucleates from wild-type (A30340), *CUP-NDC80* (A30342), *CUP-HSK3* (A31849), *CUP-NDC80 CUP-HSK3* (A31855), *CUP-CLB3* (A31847), *CUP-CLB3 CUP-NDC80* (A31853), *CUP-CLB3 CUP-HSK3* (A31851) and *CUP-CLB3 CUP-NDC80 CUP-HSK3* (A31857) (early induction = 2:15 hr, late induction= 4:30 hr after induction of sporulation; release from *NDT80* block at 4:30 hr) (n = 100). For (**F**), using a chi-square test (df 1) the fraction of binucleates with a reductional or equational division was compared between *CUP-CLB3* and *CUP-CLB3 CUP-NDC80 CUP-HSK3* $\chi^2$ = 22.28, p<0.0001. For (**G**), using a chi-square test (df 1) the fraction of binucleates with a reductional or equational division was compared between *CUP-CLB3* and *CUP-CLB3 CUP-NDC80 CUP-HSK3* $\chi^2$ = 102.7, p<0.0001.

coorientation factors applies to the conclusion that it is Clb3-CDK mediated premature microtubule–kinetochore interactions that lead to loss of centromeric cohesin protection in *CUP-CLB3* cells; disrupting microtubule–kinetochore interactions by various means restores stepwise loss of cohesin in *CUP-CLB3* cells. A simple interpretation of this result is that the centromeric-cohesin protective domain can be disrupted by tension between sister kinetochores at any meiotic stage prior to anaphase I. This does not appear to be the case. In cells lacking the coorientation factor *MAM1*, sister kinetochores come under tension in metaphase I, yet in these cells centromeric cohesin is not removed prematurely (*Toth et al., 2000* and *Figure 4—figure supplement 3*). Thus, the timing of microtubule–kinetochore interactions is of importance. It is tempting to speculate that the establishment of the centromeric-cohesin protective domain, which occurs during prophase I or perhaps even earlier, is sensitive to premature microtubule–kinetochore interactions and/or tension that promote biorientation of sister kinetochores. However, once this domain is established, its maintenance during meiosis I can no longer be disrupted by tension between sister kinetochores.

How premature microtubule–kinetochore interactions affect the centromeric cohesin protection machinery is not yet known. A defect in localization of the protective machinery to kinetochores does not appear to be the cause of this defect. Sgo1 and PP2A localize normally to kinetochores in *CUP-CLB3* cells. Therefore, lack of cohesin protection upon premature microtubule–kinetochore engagement must either result from a defect in an unknown cohesin protection pathway or from a decrease in the activity of Sgo1 and/or PP2A. Premature association of kinetochores with microtubules could result in the untimely recruitment of a factor (e.g. Clb-CDKs themselves) to the pericentromere that inhibits the cohesin protective machinery. Alternatively, microtubule–kinetochore engagement could directly affect the activity of the protective machinery. Two mechanisms have been previously proposed whereby tension modulates the activity of the cohesin protective machinery. In mammalian cells, tension spatially separates centromeric cohesin from Sgo1-PP2A, perhaps leading to loss of protection (*Lee et al., 2008*). Tension has also been proposed to cause a deformation in PP2A, thus inhibiting its catalytic activity (*Grinthal et al., 2010*). Irrespective of whether it is tension-dependent perturbation of Sgo1-PP2A and/or recruitment of inhibitory factors, it is clear that premature microtubule–kinetochore engagement is a bona fide modulator of the cohesin protective machinery.

## Regulated kinetochore assembly contributes to preventing microtubule–kinetochore interactions

Cyclin-CDKs regulate multiple aspects of microtubule–kinetochore dynamics. Cyclin-CDKs promote centrosome separation and bipolar spindle assembly (*Fitch et al., 1992*), kinetochore maturation (*Holt et al., 2009*) and chromosomal passenger complex localization (*Tsukahara et al., 2010*). Given the importance of preventing premature microtubule–kinetochore engagement to meiosis I chromosome morphogenesis, it is not surprising that cyclin-CDK activity is regulated at multiple levels in budding yeast; transcription of *CLB1*, *CLB3* and *CLB4* is not activated until cells exit pachytene (*Chu and Herskowitz, 1998*) and *CLB3* translation is restricted to meiosis II (*Carlile and Amon, 2008*).

Cyclin-CDK activity is also tightly regulated in other eukaryotes. Metazoan oocytes arrest for an extended period of time in prophase I. Multiple mechanisms keep cyclin-CDK activity low to maintain this arrest (reviewed in *Von Stetina and Orr-Weaver, 2011*). Similar regulation is observed in *D. melanogaster* and *C. elegans*. Remarkably, inappropriate activation of *Cyclin A* or cyclin E during prophase I in fruit flies and worms, respectively, results in a mitosis-like division (*Sugimura and Lilly, 2006*; *Biedermann et al., 2009*). Thus, restricting cyclin-CDK activity during premeiotic S phase and prophase I also appears to be required to establish a meiosis I-specific chromosome architecture in higher eukaryotes.

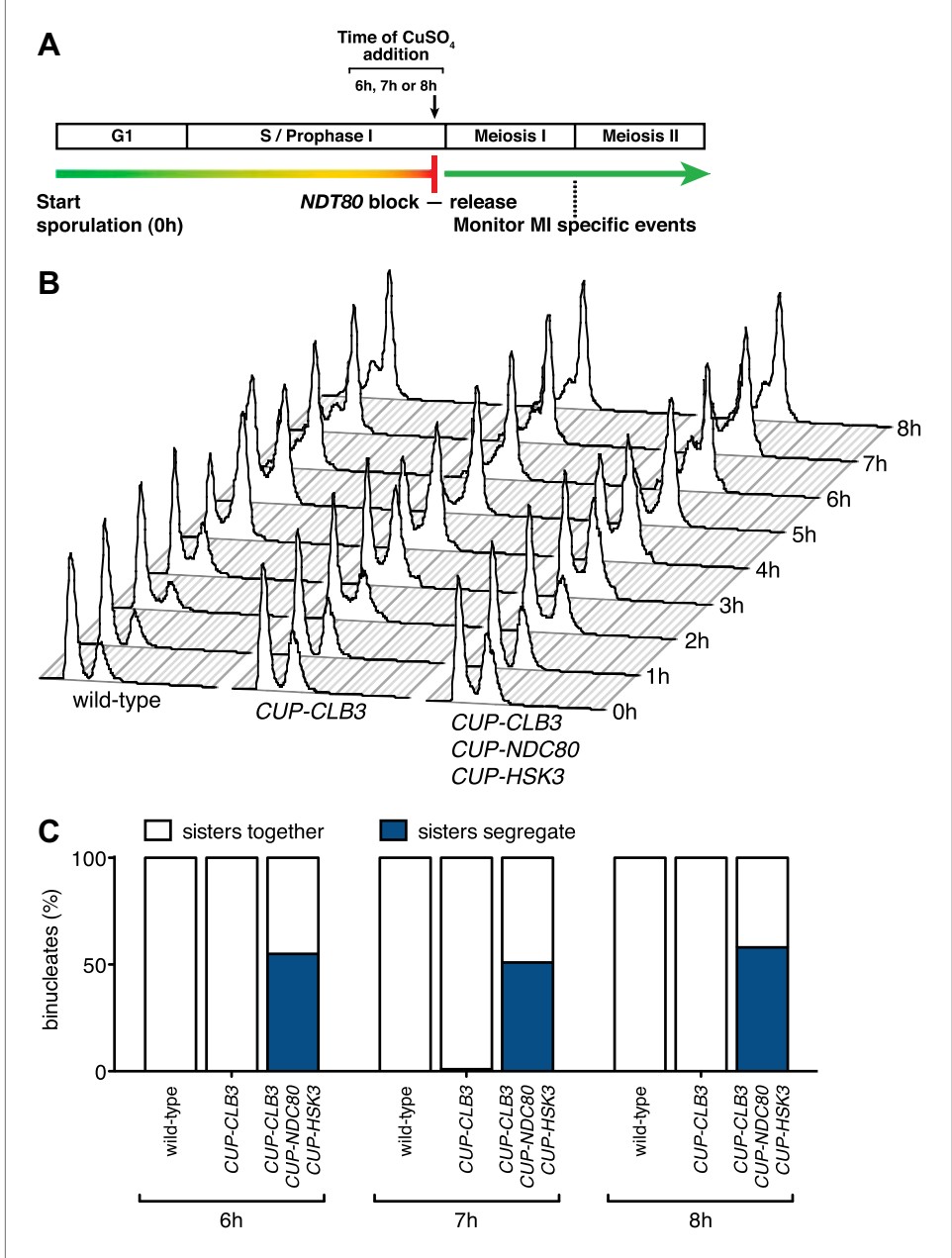

**Figure 9**. *CUP-CLB3 CUP-NDC80 CUP-HSK3*-induced meiosis I sister chromatid segregation is independent of the length of the prophase I arrest. (**A**) Schematic description of the experimental regime used in (**B**) and (**C**). (**B**), (**C**) Wild-type (A22678), *CUP-CLB3* (A22702) and *CUP-CLB3 CUP-NDC80 CUP-HSK3* (A31857) cells also carrying the *GAL4-ER* and *GAL-NDT80* fusions were induced to sporulate. Cells were released from the *NDT80* block and concurrently *pCUP1*-dependent expression was induced at either 6 hr, 7 hr or 8 hr post transfer to sporulation medium (by addition of 1 μM estradiol and 50 μM CuSO$_4$ respectively). Samples were taken at the indicated time points to determine DNA content (**B**) and the percentage of binucleate cells with segregated sister chromatids (**C**). For (**C**), using a chi-square test (df 1), the fraction of binucleates that display a reductional or equational division in *CUP-CLB3 CUP-NDC80 CUP-HSK3* cells was compared between 6 hr and 7 hr induction $\chi^2$ = 0.3212, p=0.5709 and between 6 hr and 8 hr induction $\chi^2$ = 0.1831, p=0.6687.

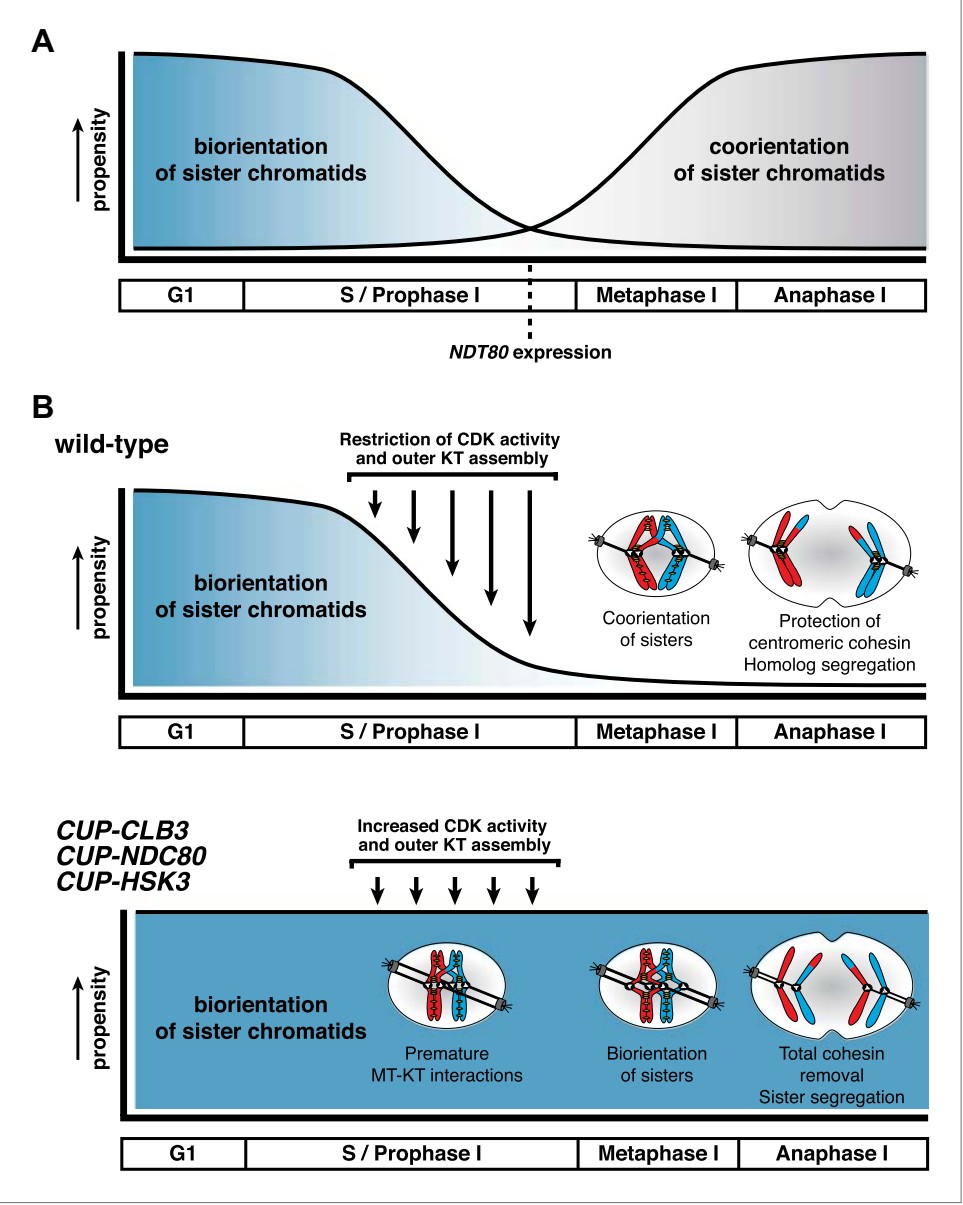

**Figure 10**. Model for temporal regulation of microtubule–kinetochore interactions during meiosis. (**A**) As prophase I progresses, the propensity of sister chromatids to biorient decreases and the ability to coorient sister chromatids increases. (**B**) Top panel: inhibiting Clb-CDK activity and outer kinetochore (KT) assembly during prophase I establishes a meiosis I-specific chromosome segregation pattern by allowing sister kinetochore coorientation and protection of centromeric cohesin. Bottom panel: disrupting the regulation of microtubule–kinetochore (MT–KT) interactions causes sister chromatid segregation in meiosis I.

Restriction of cyclin-CDK activity during premeiotic S phase and prophase I appears to be the major mechanism preventing premature microtubule–kinetochore interactions, but our data indicate that regulation of outer kinetochore assembly serves as an additional mechanism to prevent this from occurring. *CUP-CLB3* can only induce meiosis I sister chromatid segregation when expressed during premeiotic S phase/early prophase I, but fails to do so when expressed during late prophase I. This difference is likely due to the outer kinetochore being present only until early prophase I. When Ndc80, Hsk3 and Clb3 are coexpressed during late prophase I, sister chromatid segregation occurs in meiosis I. This result demonstrates that the presence of Clb3-CDKs alone during late prophase I is not sufficient to cause meiosis I sister chromatid segregation but that outer kinetochore

components must also be expressed. Whether outer kinetochore disassembly solely occurs to prevent microtubule kinetochore interactions remains to be determined. Outer kinetochore disassembly could also serve additional purposes during prophase I such as enabling telomere-mediated chromosome movements. Further study of the kinetochore assembly/disassembly cycle during meiosis will provide insights into the full impact of kinetochore regulation on meiotic chromosome segregation.

In budding yeast, two essential components of the outer kinetochore, Ndc80 and Hsk3, are downregulated during prophase I. In *S. pombe*, Ndc80 and its binding partner Nuf2 dissociate from kinetochores in prophase I (*Asakawa et al., 2005*) raising the interesting possibility that deconstruction of the outer kinetochore is a conserved feature of meiotic prophase I. This dissociation depends on the mating pheromone signaling pathway (*Asakawa et al., 2005*). Intriguingly, ectopic induction of meiosis without mating pheromone signaling (i.e. in *pat1* mutants), results in segregation of sister chromatids instead of homologous chromosomes in meiosis I (*Yamamoto and Hiraoka, 2003*; *Yamamoto et al., 2004*). Perhaps this change in the pattern of chromosome segregation in *pat1* mutants arises from premature microtubule–kinetochore interactions due to a defect in outer kinetochore disassembly. Interestingly, in mouse oocytes, the Ndc80 complex is recruited to chromosomes only after nuclear envelope breakdown (*Sun et al., 2011*), raising the possibility that outer kinetochore assembly is also prevented in meiotic prophase I in vertebrates.

## Concluding remarks

Proper segregation of the genome during gametogenesis is critical for the proliferation of sexually reproducing species. Errors in chromosome segregation during meiosis result in aneuploidy, the leading cause of birth defects and miscarriages in humans (*Hassold and Hunt, 2001*). Thus, it is crucial to understand how accurate meiotic chromosome segregation is achieved. We discovered that the establishment of a meiosis-specific chromosome segregation pattern depends on the regulation of microtubule–kinetochore interactions. This is achieved by regulating cyclin-CDK activity as well as assembly of the outer kinetochore. There is evidence for similar regulatory events across different organisms (*Asakawa et al., 2005*; *Sugimura and Lilly, 2006*; *Biedermann et al., 2009*; *Von Stetina and Orr-Weaver, 2011*), suggesting that temporal restriction of microtubule-kinetochore interactions is an evolutionarily conserved event required to execute proper meiotic chromosome segregation.

# Materials and methods

## Strains and plasmids

Strains used in this study are described in *Supplementary file 1* and are derivatives of SK1 (all meiosis experiments) or W303 (*Figure 5A*). GAL-NDT80 and GAL4-ER constructs are described in *Benjamin et al. (2003)*. CUP-CLB1, CUP-CLB3, CUP-CLB4, CUP-CLB5, SPC42-mCherry, SGO1-3V5, RTS1-13myc, RTS1-3V5, HSK3-3V5, NDC80-3V5, ASK1-13myc, CUP-NDC80-3V5, CUP-HSK3, mam1Δ, SPO13-3V5, mad3Δ, DAM1-3V5, CUP-HSK3-3HA were constructed by PCR-based methods described in *Longtine et al. (1998)*. Primer sequences for strain constructions are available upon request. ndc80-1 and dam1-1 are described in *Wigge et al. (1998)*; *Jones et al. (1999)* and SK1 strains carrying these alleles were constructed via backcrossing (>9X). CENV-LacO was constructed by cloning a CENV homology region with XhoI restriction sites into the SalI cut plasmid pCM40 (gift from Doug Koshland) and integrated near *CDEIII* (<1 kb) by BamHI digest. pHG40 carrying *CUP1* promoter was a gift from Hong-Guo Yu. 3V5 tagging plasmids were provided by Vincent Guacci.

## Sporulation conditions

Strains were grown to saturation in YPD at room temperature, diluted in BYTA (1% yeast extract, 2% tryptone, 1% potassium acetate, 50 mM potassium phthalate) to $OD_{600} = 0.25$, and grown overnight at 30°C (room temperature for *ndc80-1* and *dam1-1* experiments). Cells were resuspended in sporulation medium (0.3% potassium acetate [pH 7], 0.02% raffinose) to $OD_{600} = 1.85$ and sporulated at 30°C unless otherwise indicated. *GAL-NDT80 GAL4-ER* strains were released from the *NDT80* block by the addition of 1 μM β-estradiol (5 mM stock in ethanol; Sigma E2758-1G, St. Louis, MO) at 4 hr 30 min unless otherwise indicated. Note: strains released from *NDT80* block at 4 hr 30 min are prototrophic and have accelerated meiotic kinetics relative to strains containing auxotrophies. Strains with *CUP1* promoter

driven alleles were induced by addition of $CuSO_4$ (50 µM final concentration; 100 mM stock made from anhydrous powder [FW = 159.6 g/mol]; Mallinckrodt, Hazelwood, MO) at indicated times.

## Transient inactivation of the *ndc80-1* or *dam1-1* alleles

Wild-type, *ndc80-1* or *dam1-1* cells carrying *GAL-NDT80 GAL4-ER* were induced to sporulate at room temperature (permissive temperature). After 2 hr 45 min, cyclin expression was induced by addition of 50 µM $CuSO_4$ and cells were concurrently shifted to the semi-permissive (34°C) or non-permissive (>35.5°C) temperature and allowed to arrest in pachytene. Cells were then transferred to the permissive temperature and released from the *NDT80* block by addition of 1 µM β-estradiol into either a metaphase I arrest (by depleting Cdc20) or allowed to proceed through the meiotic divisions.

## Benomyl treatment of meiotic cultures

Wild-type or *CUP-CLB3* cells carrying the *GAL-NDT80 GAL4-ER* constructs were induced to sporulate at 30°C. 2 hr 15 min after transfer into sporulation medium, cells were filtered and transferred to medium containing $CuSO_4$ (50 µM) and either 0.4% DMSO or benomyl (120 µg/ml). After an additional 2 hr 15 min incubation, benomyl was washed out by filtering and washing cells with 10 volumes of sterile $dH_2O$ containing 0.4% DMSO. Cells were subsequently resuspended in sporulation medium containing 1 µM β-estradiol to release from *NDT80* block. The efficacy of benomyl treatment was confirmed by spindle morphology. See *Hochwagen et al. (2005)* for further technical details regarding benomyl resuspension in sporulation medium.

## Mitotic induction of monopolin

*MAT***a** haploid cells carrying the *MET-CDC20* or *MET-CDC20 GAL-CDC5 GAL-MAM1* fusions and CENIV-GFP dots cultured in complete synthetic medium without methionine (CSM-MET) containing 2% raffinose were arrested in G1 with 5 µg/ml α-factor. For *Figure 5A* condition (1), cells were treated with galactose (to induce Cdc5 and Mam1 production) for 1 hr prior to α-factor release. When arrest was complete, cells were released into rich medium (YEP) with 2% raffinose lacking pheromone and containing 2% galactose, 1% DMSO and 8 mM methionine (to repress Cdc20 production). 8 mM methionine was added every hour to maintain metaphase arrest. When metaphase arrest was complete, cells were released into CSM-MET medium, containing 2% dextrose, 1% DMSO and 5 µg/ml α-factor. For condition (2), G1 arrested cells were released into YEP medium with 2% raffinose, lacking pheromone, containing 8 mM methionine and 1% DMSO. 8 mM methionine was added every hour to maintain the metaphase arrest. After 2 hr, cells were treated with 2% galactose for 1 hr and were subsequently released into CSM-MET medium, containing 2% dextrose, 1% DMSO and 5 µg/ml α-factor. For condition (3), G1 arrested cells were released into YEP medium with 2% raffinose, lacking pheromone, containing 8 mM methionine and 15 µg/ml nocodazole in DMSO. 8 mM methionine was added every hour to maintain the metaphase arrest. After 2 hr, cells were treated with 2% galactose for 1 hr and were subsequently released into CSM-MET medium, containing 2% dextrose, 1% DMSO and 5 µg/ml α-factor. Samples were taken every 15 min after release from metaphase arrest to determine GFP dot segregation in anaphase.

## Indirect immunofluorescence

Indirect immunofluorescence was performed as described in *Kilmartin and Adams (1984)*. Spindle morphologies were classified as follows: metaphase I or metaphase I-like spindles were defined as a short, bipolar spindle spanning a single DAPI mass. Anaphase I spindles were defined as an elongated spindle spanning two distinct DAPI masses. Metaphase II spindles were defined as two short, bipolar spindles, each spanning a DAPI mass. Anaphase II spindles were defined as two elongated spindles, each spanning two distinct DAPI masses (four DAPI masses total). For *Figure 8D,E*, robust bipolar spindle was classified as a short, thick, bipolar spindle with equal intensity tubulin staining across the entire length of the spindle. A fragile spindle was classified as a short bipolar spindle with lower intensity tubulin staining in the middle of the spindle axis.

## Live cell imaging

Cells were induced to sporulate and $CuSO_4$ was added at the indicated times. After 30–60 min post $CuSO_4$ induction, cells were layered on a Concanavalin A (2 mg/ml; stock solution 20 mg/ml diluted in 50 mM $CaCl_2$, 50 mM $MnSO_4$) coated cover slip and assembled into an FCS2 fluidic chamber (Bioptechs

Inc. Butler, PA). Sporulation medium was heated to 30°C, aerated using an aquarium air pump (Petco Animal Supplies, Inc. Cambridge, MA) and was perfused into the fluidic chamber using a peristaltic pump (Gilson Inc., Middleton, WI) with a flow rate of 4–7 ml/h. Alternatively, cells were induced to sporulate as above and transferred to a microfluidic chamber (CellASIC Corp. Hayward, CA). Cells were imaged using a Zeiss Axio Observer-Z1 with a 100× objective (NA = 1.45), equipped with a Hamamatsu ORCA-ER digital camera. 11 Z-stacks (1 micron apart) were acquired and maximally projected. Metamorph software was used for image acquisition and processing. Images for *Figure 2B* was processed using Metamorph deconvolution software. For *Figure 2C*, a cell was scored as harboring a separated pair of sister kinetochores if the heterozygous CENV-GFP dot signal underwent transient splitting for at least two time points for the duration of the movie.

## GFP-dot and Spc42-mCherry cell fixation conditions

An aliquot of cells was fixed with 3.7% formaldehyde in 100 mM phosphate buffer (pH 6.4) for 10–15 min. Cells were washed once with 100 mM phosphate, 1.2 M sorbitol buffer (pH 7.5) and permeabilized with 1% Triton X-100 stained with 0.05 µg/ml 4′,6-diamidino-2-phenylindole (DAPI). Cells were imaged using a Zeiss Axioplan 2 microscope or a Zeiss Axio Observer-Z1 with a 100× objective (NA = 1.45), equipped with a Hamamatsu ORCA-ER digital camera. Openlab or Metamorph software was used for image acquisition and processing.

## Chromosome spreads

4 $OD_{600}$ units of cells were harvested and spheroplasted with 0.1 mg/ml zymolyase 100T (Seikagaku Corp, Japan) and 15 mM DTT in solution 1 (2% potassium acetate, 0.8% sorbitol) for 10–13 min at 37°C. Ice-cold solution 2 (100 mM MES [pH 6.4], 1 mM EDTA, 0.5 mM $MgCl_2$, 1 M sorbitol) was added to stop spheroplasting and cells were centrifuged at 2500 rpm for 2–3 min. The supernatant was discarded and the pellet was gently resuspended in 100–200 µl of solution 2. 15 µl of the resuspension was spread onto a glass slide. Subsequently, 30 µl of fixative solution (4% paraformaldehyde, 3.4% sucrose), 60 µl of 1% lipsol and 60 µl of fixative solution were added on top of cell suspension and spread using a glass rod seven to ten times back and forth. The slides were dried for at least 2 hr at room temperature, rehydrated in PBS pH 7.4, blocked with 0.2% gelatin, 0.5% BSA in PBS, and stained as described in the 'Antibody' section. For quantifications of spread nuclei, images were acquired using a Zeiss Axioplan 2 microscope or a Zeiss Axio Observer-Z1 with a 100× objective (NA = 1.45), equipped with a Hamamatsu ORCA-ER digital camera. Openlab or Metamorph software was used for image acquisition and processing. 40–100 spread nuclei were counted for each sample, except for strain A31955 in *Figure 8B* (n = 28). Two proteins were identified as colocalized in spread nuclei when more than 90% of foci overlapped. They were defined as partially colocalized when the overlap between foci was approximately 50% and as mislocalized when the overlap was negligible.

## In vitro kinase assay

In vitro kinase assays were performed as described in *Carlile and Amon (2008)* with the following modifications: 1 mg of total protein was incubated with 40 µl of 50% slurry anti-V5 agarose affinity gel (Sigma, St. Louis, MO) for 2 hr at 4°C. One half of the immunoprecipitate was used for the in vitro kinase assay, while the other half was used for Western blotting to detect Cdc28-3V5.

## Western blot analysis

For immunoblot analysis, ~10 $OD_{600}$ units of cells were harvested and treated with 5% trichloroacetic acid for at least 10 min at 4°C. The acid was washed away with acetone and the cell pellet was subsequently dried. The cell pellet was pulverized with glass beads in 100 µL of lysis buffer (50 mM Tris–HCl at pH 7.5, 1 mM EDTA, 2.75 mM DTT, complete protease inhibitor cocktail [Roche, Basel, Switzerland]) using a bead-beater (Biospec Products, Inc. Bartlesville, OK). 3× SDS sample buffer was added and the cell homogenates were boiled. Standard procedures for sodium dodecyl sulfate–polyacrylamide gel electrophoresis (SDS-PAGE) and Western blotting were followed (*Laemmli, 1970*; *Towbin et al., 1979*; *Burnette, 1981*). A nitrocellulose membrane (VWR International LLC, Radnor, PA) was used to transfer proteins from polyacrylamide gels. Antibody dilutions are described in the 'Antibody' section.

## Flow cytometry

1 ml aliquot of a meiotic culture was spun down and the pellet was re-suspended in 70% ethanol and fixed for at least 60 min. Ethanol was removed and the cell pellet was washed with 50 mM

sodium citrate, pH 7 and sonicated for 6 s at 50% output. The sample was subsequently incubated with 0.25 mg/ml Ribonuclease A (Sigma, St. Louis, MO) in 50 mM sodium citrate overnight at 37°C, washed once with 50 mM sodium citrate and re-suspended in 50 mM sodium citrate with either 1 µM Sytox Green (Molecular Probes, Carlsbad, CA) or 16 µg/ml propidium iodide (Sigma). Samples were analyzed using FACSCalibur (Becton Dickenson Co. Franklin Lakes, NJ).

## Chromatin immunoprecipitation

400 $OD_{600}$ units of cells were fixed for 15 min at room temperature in 1% formaldehyde. The formaldehyde was quenched by addition of 125 mM glycine. Samples were processed as previously described (*Vader et al., 2011*). Before immunoprecipitation, 120th of the sample was removed as the input sample. The antibodies used for immunoprecipitation are described in the 'Antibody' section. For ChIP-chip, samples were processed and analyzed as described in *Vader et al. (2011)*. For qPCR analysis, DNA was amplified using SYBR Premix ExTaq Perfect Real Time Kit (Takara Bio Inc. Otsu, Shiga, Japan). PCR reactions were 40 cycles of 95°C, 20 s; 55°C, 30 s; 72°C, 30 s using a Roche LightCycler 480 II (Roche, Basel, Switzerland). The following primers were used (5'–3'):

**CENV F:** CTT GTT TAG TGC AAG CCA CTG TT
**CENV R:** CCG CAT TTC CTT GAT TTA CTG TC
**c281 F:** CAA CGA ACC GTG GGA ACG TTA TAG
**c281 R:** GAA ACT TTC CTG GTA CCT TCT GC
**c194 F:** GCT GAA AGC ATG CCA CTG TA
**c194 R:** GGT GTT CCT GCT TCG TTG TTA G
**HMR F:** ACG ATC CCC GTC CAA GTT ATG
**HMR R:** CTT CAA AGG AGT CTT AAT TTC CCT G

## Recombination southern

~20 $OD_{600}$ units of cells were harvested and treated with sodium azide (0.1% final concentration). Cells were pelleted and snap frozen in liquid nitrogen. Genomic DNA was extracted as follows: Cells were washed once in TE and spheroplasted with 1/100 volume of beta-mercaptoethanol and 250 µg/ml zymolyase T100 in spheroplasting buffer (1 M sorbitol, 42 mM $K_2HPO_4$, 8 mM $KH_2PO_4$, 5 mM EDTA) for 30 min at 37°C on a rotating rack. 100 µl preheated (65°C) lysis buffer (1:1 mix of 1 M Tris pH 8 and 0.5 M EDTA, 2.5–3% SDS) was added and mixed by inverting. 15 µl proteinase K (18 ± 4 mg/ml PCR grade solution; Roche, Basel, Switzerland) was added and incubated at 65°C for ~1.5 hr. Subsequently, 150 µl 5 M potassium acetate was added, mixed by inverting and transferred to 4°C for 10 min. Samples were centrifuged at 4°C for 20 min and 650 µl of supernatant was transferred into a 2 ml tube containing 750 µl 100% ethanol, avoiding as much of the white fluff as possible. Samples were mixed by inverting and left at 4°C for 10 min. Nucleic acid was precipitated at 15,000 rpm for 10 min, 4°C. Samples were subsequently resuspended in TE and treated with RNase A (50 µg/ml; Roche), for 15–20 min at 37°C and kept at 4°C overnight. DNA was extracted with phenol/chloroform/isopropanol and was resuspended in 125 µl TE. XhoI-MluI digested DNA fragments were separated on 0.6% agarose gel in 1× TBE and transferred onto Hybond-XL plus membranes (GE Healthcare Biosciences, Pittsburgh, PA) by alkaline transfer. Southern blotting was performed as previously described (*Hunter and Kleckner, 2001*).

## Antibodies

### Indirect immunofluorescence

Spindle morphology was determined using a rat anti-tubulin antibody (Serotec, Kidlington, UK) used at a dilution of 1:100, and anti-rat FITC antibodies (Jackson ImmunoResearch Laboratories, Inc. West Grove, PA) used at a dilution of 1:100–200.

### Western blotting

Lrs4-13myc, Rec8-13myc, Ask1-13myc and Mam1-9myc were detected using a mouse anti-myc antibody (Covance, Princeton, NJ) at a 1:500 dilution. Rec8-3HA and Hsk3-3HA were detected using a mouse anti-HA antibody (HA.11; Covance) at a 1:1000 dilution. Hsk3-3V5, Ndc80-3V5 and Dam1-3V5 were detected using a mouse anti-V5 antibody (Invitrogen, Carlsbad, CA) at a 1:2000 dilution. Pgk1 was detected using a mouse anti-Pgk1 antibody (Molecular Probes, Carlsbad, CA) at a

1:10,000 dilution. Clb3 was detected using a rabbit anti-Clb3 antibody (Sc7167; Santa Cruz Biotechnology Inc. Santa Cruz, CA) at a 1:500 dilution. Kar2 was detected using a rabbit anti-Kar2 antibody (kindly provided by Mark Rose) at a 1:200,000 dilution. The secondary antibodies used were a sheep anti-mouse antibody conjugated to horseradish peroxidase (HRP) (GE Healthcare Biosciences, Pittsburgh, PA) at a 1:5000 dilution or a goat anti-rabbit antibody conjugated to HRP (BioRad, Hercules, CA) at a 1:10,000 dilution. Antibodies were detected using the SuperSignal West Pico Chemiluminescent Substrate (Thermo Scientific, Waltham, MA).

## Chromatin immunoprecipitation

Rec8-3HA was immunoprecipitated using 2–5 μg of rat anti-HA antibody (3F10; Roche, Basel, Switzerland) in combination with 50 μl of 50% slurry Protein G beads (Roche). Rec8-13myc was immunoprecipitated using 2–5 μg of mouse anti-myc antibody (9E11) in combination with 50 μl of 50% slurry Protein G beads (Roche). Sgo1-3V5 and Spo13-3V5 were immunoprecipitated with 40–50 μl of 50% slurry anti-V5 agarose affinity gel (Sigma, St. Louis, MO). Pds5 was immunoprecipitated using 1.3μl of rabbit anti-Pds5 antibody (kindly provided by Vincent Guacci) in combination with 50 μl of 50% slurry Protein A beads (Roche). Phosphorylated Rec8 was immunoprecipitated using 2 μg of rabbit anti-phospho-S179 Rec8 or rabbit anti-phospho-S521 Rec8 in combination with 50 μl of 50% slurry Protein A beads (Roche).

## Chromosome spreads

Lrs4-13myc, Ndc10-13myc, Sgo1-9myc, Rts1-13myc, Rec8-13myc, Ask1-13myc, and Mam1-9myc were detected using a preabsorbed rabbit anti-myc antibody (Gramsch, Schwabhausen, Germany) at a 1:400 dilution. Ndc10-6HA and Rec8-3HA were detected using either a preabsorbed mouse anti-HA antibody (HA.11; Covance, Princeton, NJ) or a rat anti-HA antibody (3F10; Roche, Basel, Switzerland) at a 1:400 dilution. Ndc80-3V5 was detected using a mouse anti-V5 antibody (Invitrogen, Carlsbad, CA) at a 1:400 dilution. Zip1 was detected using y-300 rabbit antibody (Santa Cruz Biotechnology Inc. Santa Cruz, CA) at a 1:400 dilution. Rad51 was detected using y-180 rabbit IgG (Santa Cruz Biotechnology Inc.) at a 1:400 dilution. Secondary antibodies used were preabsorbed anti-rabbit FITC antibody (Jackson ImmunoResearch Laboratories, Inc. West Grove, PA), preabsorbed anti-rat CY3 antibody (Jackson ImmunoResearch Laboratories, Inc.) or preabsorbed anti-mouse CY3 antibody (Jackson ImmunoResearch Laboratories, Inc.) at a 1:400–1:800 dilution.

## Cluster analysis and ordered plots for mRNA-seq and ribosome footprinting data

Cluster analysis of the ribosome footprinting data for the kinetochore components listed in *Figure 6—figure supplement 1* was performed using Cluster 3.0. Genes were clustered by hierarchical average based on Spearman correlation using mean centered arrays. Clustering data (*Figure 6B*, *Figure 6—figure supplement 2*) were visualized using Java Treeview. Note that ribosome footprints are normalized such that the sum of expression across the time course is equivalent for each gene. For plots in *Figure 6C,D* and *Figure 6—figure supplements 3–10*, mRNA-seq and ribosome footprinting data were plotted for indicated genes based on the dataset from *Brar et al. (2012)*. The meiotic stages plotted on the x-axis are in the following order: vegetative (gb15 exponential and A14201 exponential), meiotic entry (1, A, B and D), DNA replication (E and F), recombination (G and I), prophase I (3 and 4), metaphase I (5 and 6), anaphase I (7 and 8), metaphase II (9 and 10), anaphase II (11, 12 and 13) and spore formation (15 and 18). The detailed explanation of the above letter and number codes can be found in *Brar et al. (2012)*.

## Statistical analysis

Chi-square ($\chi^2$) tests were performed using GraphPad Prism 6.0 software with two-tailed P values and 95% confidence intervals. Corresponding degrees of freedom (df), $\chi^2$ and P values are shown in the figure legends.

## Acknowledgements

We are grateful to Hong-Guo Yu, Vincent Guacci, and Wolfgang Zachariae for reagents, Hannah Blitzblau, Gerben Vader, Jingxun Chen, Kristin Kuhn, Kristin Knouse, Ann Thompson, André and Charles Felts for technical assistance, Steve Bell, Leon Chan, Dean Dawson, Doug Koshland, Andrew Murray, Terry Orr-Weaver and members of the Amon lab for their critical reading of this manuscript.

# Additional information

## Funding

| Funder | Grant reference number | Author |
|---|---|---|
| Howard Hughes Medical Institute | | Matthew P Miller, Elçin Ünal, Gloria A Brar, Angelika Amon |
| National Institutes of Health | GM62207 | Matthew P Miller, Elçin Ünal, Gloria A Brar, Angelika Amon |
| Jane Coffin Childs Memorial Fund | | Elçin Ünal |
| American Cancer Society | | Gloria A Brar |

The funders had no role in study design, data collection and interpretation, or the decision to submit the work for publication.

## Author contributions

MPM, Conception and design, Acquisition of data, Analysis and interpretation of data, Drafting or revising the article; EÜ, Conception and design, Acquisition of data, Analysis and interpretation of data, Drafting or revising the article; GAB, Acquisition of data, Analysis and interpretation of data, Drafting or revising the article; AA, Conception and design, Analysis and interpretation of data, Drafting or revising the article

# Additional files

## Supplementary files
• Supplementary file 1. Strains used in this study.

## Major datasets

The following datasets were generated

| Author(s) | Year | Dataset title | Dataset ID and/or URL | Database, license, and accessibility information |
|---|---|---|---|---|
| Miller MP, Ünal E, Brar GA, Amon A | 2012 | Mapping of cohesion factor association sites in *S. cerevisiae*—Meiosis I chromosome segregation is established through regulation of microtubule–kinetochore interactions | GSE41339; http://www.ncbi.nlm.nih.gov/geo/query/acc.cgi?acc=GSE41339 | Publicly available at GEO |

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
