## [Decision Letter]

Thank you for choosing to send your work entitled “Meiosis I Chromosome Segregation is Established by Inhibiting Microtubule-Kinetochore Interactions in Prophase I” for consideration at *eLife*. Your article has been evaluated by a Senior Editor and 3 reviewers, one of whom is a member of *eLife's* Board of Reviewing Editors.

The Reviewing Editor and the other reviewers discussed their comments before we reached this decision, and the Reviewing Editor has assembled the following comments based on the reviewers' reports. Our goal is to provide the essential revision requirements as a single set of instructions, so that you have a clear view of the revisions that are necessary for us to publish your work.

**General assessment and substantive concerns to be addressed during revision:**

Amon and colleagues previously showed that aberrant expression of Clb3 in meiotic prophase I induces sister segregation in the first meiotic division. Here, Miller et al. show that this is due to early development of bipolar spindles and present evidence that premature kinetochore-microtubule attachments disrupt normal morphogenesis of kinetochores and pericentric cohesion. That these connections must be at least transiently disrupted to allow development of monopolar kinetochores and in particular a protective cohesin domain formed around the centromeres is an important and new discovery. This is a typical tour de force effort from the Amon lab, with cleverly designed and well-executed experiments and several novel observations. The data are for the most part impeccable. However, the simple message is perhaps lost by emphasis on other points that are either overstated or speculative without careful comment to these issues. Furthermore, certain novelty claims are unwarranted and need rewriting (see point 1). Substantial points that must be addressed in a revision are given below in no particular order with regard to significance.

1. The paper of Asakawa et al. (MBC 2005) should be discussed earlier and more comprehensively. Presenting this information in the discussion section as it now reads is confusing. This paper demonstrated meiosis-specific disassembly of the outer kinetochore (the *S. pombe* Nif2-Ndc80 complex) and indicated that this is required for reductional segregation at MI. Moreover, Asakawa also showed that this process requires the Pat1 kinase, the Mei2 regulator of meiotic transcript stability, and mating-pheromone signaling. The current study in this regard is an advance in important ways, that this is critical in part for connections, but does not supersede it.

2. It's surprising that there is no figure at the beginning of the Results showing westerns of CLB overexpression. It is important to see timing and know how the overexpression compares to the levels that normally form in WT. There are such figures for Clb3 later, e.g., in panels of Fig 2B and Fig S2B, but there is no explicit discussion of what's seen with respect to timing or levels, and the other CLBs are not shown. It would be interesting to know if the differences in phenotypes detected for the specific cyclins were because of kinase protein levels or activities? One of the most interesting points developed in the present story was that Clb4 expression was only capable of disrupting SPB but not cohesion – is this specificity or levels?

3. Most of the data are superb, but the recombination Southern and quantification are not (fig S1E): FigureS1E shows that recombination is not normal in GAL-CLB3 cells - the crossovers form earlier (at least 1 hr and with biphasic kinetics) and at lower levels than seen in wild type. However, the authors claim that recombination is normal when CLB3 is prematurely expressed. This is a contradiction that needs to be addressed. In fact, it is hard to believe that the quantification is accurate since it indicates that products are showing up at high levels already at 3 hr in the GAL-CLB3 strain, even though no products are obvious on the Southern at this time point. Also, it seems a bit odd to include DSBs along with recombinant DNA molecules as “recombination products”.

4. We are concerned about the physiological relevance of a number of experiments. While the phenomena described here are fascinating, how relevant is the pathology associated with very early, very high expression of Clb3 for understanding wild-type meiosis? This needs some discussion and balance with regard to proof versus speculation. The introduction section does not make a link to how over-expression might be used to learn about the normal process. Further some conclusions remain speculative with the approach and should be stated as such.

A) For example, with respect to the physiological role of outer-kinetochore disassembly during prophase, the authors' interpretation is that this process is redundant with suppression of cyclins in preventing premature kinetochore-spindle connections. But it seems that prevention of prophase spindle assembly completely overrides any need to disassemble the outer kinetochore; bipolar spindles never form in prophase I (Figure 1).

B) Moreover, artificial assembly of the outer kinetochore in prophase I (CUP-NDC80 CUP-HSK3) does not influence MI segregation unless CLB3 is also artificially expressed. This is an important paradox that must be addressed. It remains completely unproven whether the physiological role of outer kinetochore disassembly in prophase I is to prevent premature spindle-kinetochore connections. It seems equally possible that SPB-kinetochore connections must be dissociated for normal execution of prophase I events, e.g. telomere-nuclear envelope attachment, chromosome movement, etc.

5. For most experiments, CUP-CLB3 is induced so early that cells may still be in S phase. FACs analysis for representative time courses should be shown. The induction of Ndt80 expression at 4.5 hrs is also questionably early - has a majority of the culture reached the ndt80 arrest point at 4.5 hrs? The concern is that early, non-physiological expression of Cdc5 is also occurring in these experiments and may actually be required for the observed phenomena. What happens if Ndt80 is expressed later? There isn't a single time-course in this paper to demonstrate when MI occurs in a typical wild-type culture. The time-course in FigureS1E shows that crossovers don't form until 7 hrs making it unlikely that cells were in mid/late pachytene at 4.5 hrs.

6. Similarly it is an overstatement to write categorically that the work shows two independent mechanisms for timely engagement of the microtubules with the kinetochore. Manipulating the synthesis of the outer kinetochore proteins enhances the Cdk effects, but perhaps this is because the unknown kinase targets are not completely modified? One might suggest that the transcription or translation of the kinetochore proteins is regulated by a repressor whose activity is ablated by Cdk function.

---

## [Author Response]

*1. The paper of Asakawa et al. (MBC 2005) should be discussed earlier and more comprehensively. Presenting this information in the discussion section as it now reads is confusing. This paper demonstrated meiosis-specific disassembly of the outer kinetochore (the S. pombe Nif2-Ndc80 complex) and indicated that this is required for reductional segregation at MI. Moreover, Asakawa also showed that this process requires the Pat1 kinase, the Mei2 regulator of meiotic transcript stability, and mating-pheromone signaling. The current study in this regard is an advance in important ways, that this is critical in part for connections, but does not supersede it*.

We have included a detailed discussion of the Asakawa paper in the Results section.

*2. It's surprising that there is no figure at the beginning of the Results showing westerns of CLB overexpression. It is important to see timing and know how the overexpression compares to the levels that normally form in WT. There are such figures for Clb3 later, e.g., in panels of Fig 2B and Fig S2B, but there is no explicit discussion of what's seen with respect to timing or levels, and the other CLBs are not shown. It would be interesting to know if the differences in phenotypes detected for the specific cyclins were because of kinase protein levels or activities? One of the most interesting points developed in the present story was that Clb4 expression was only capable of disrupting SPB but not cohesion – is this specificity or levels*?

Several points are of importance here. First, Clb3 produced from the *GAL1-10* promoter (and driven by estrogen induction) is expressed at similar levels as endogenous Clb3. This was demonstrated in Carlile and Amon (2008) and is now mentioned in the text. Thus, when Clb3 is driven from the *GAL1-10* promoter the protein is not overexpressed, just prematurely expressed. Expression from the *CUP1* promoter results in a 5-fold overexpression compared to the *GAL1-10* promoter. This is now shown in Figure 1B. Importantly, the degree of expression has no effect on the phenotype. The percentage of cells segregating sister chromatids during meiosis I is similar in *CUP-CLB3* and *GAL-CLB3* cells (shown in Figure 1C). This result indicates that degree of ectopic expression is irrelevant for the phenotype. The only aspect that matters is *premature* expression. This is now explained in detail in the text.

The reader may further wonder why we did not use the *GAL-CLB3* construct but instead used the *CUP-CLB3* fusion for most of our experiments, if the former only leads to premature expression but the latter to premature expression and overexpression. This is now also explained. Figure 1A shows that estrogen addition during early stages of sporulation interferes with meiotic progression, precluding the analysis of the meiotic products.

Finally, as requested by the reviewers we have compared protein levels and kinase activity of the various cyclins expressed from the *CUP1* promoter. Figure 1D shows that the various Clb cyclins are expressed at similar levels when driven from the *CUP1* promoter. Figure 1F shows that there is no correlation between the amount of kinase activity produced by the various cyclins expressed from the *CUP1* promoter and the ability to induce sister chromatid segregation during meiosis I. Thus cyclin specificity, not quantity, is responsible for the differences in phenotype.

*3. Most of the data are superb, but the recombination Southern and quantification are not (fig S1E): FigureS1E shows that recombination is not normal in GAL-CLB3 cells - the crossovers form earlier (at least 1 hr and with biphasic kinetics) and at lower levels than seen in wild type. However, the authors claim that recombination is normal when CLB3 is prematurely expressed. This is a contradiction that needs to be addressed. In fact, it is hard to believe that the quantification is accurate since it indicates that products are showing up at high levels already at 3 hr in the GAL-CLB3 strain, even though no products are obvious on the Southern at this time point. Also, it seems a bit odd to include DSBs along with recombinant DNA molecules as “recombination products”*.

We re-quantified the recombination products by taking the ratio of R2 over P1. The new graph displayed in Figure 2 – Figure Supplement 5 shows no significant impact of prematurely expressed *CLB3* on recombination.

*4. We are concerned about the physiological relevance of a number of experiments. While the phenomena described here are fascinating, how relevant is the pathology associated with very early, very high expression of Clb3 for understanding wild-type meiosis? This needs some discussion and balance with regard to proof versus speculation. The introduction section does not make a link to how over-expression might be used to learn about the normal process. Further some conclusions remain speculative with the approach and should be stated as such*.

The reviewers are concerned that our analyses solely rely on experiments that overexpress *CLB*s and that the “pathology” described here has no bearing on the effects of *CLB*s in wild-type meiosis. We fundamentally disagree with this assessment.

First, as described in Carlile and Amon (2008) and now explicitly mentioned in the revised manuscript, expression of *CLB3* from the *GAL1-10* promoter does not lead to non-physiological levels of Clb3. The levels produced from the *GAL1-10* promoter during meiosis I are the same as observed in wild-type cells during meiosis II. More importantly, Figure 1C shows that amount of premature Clb3 expression does not impact the phenotype. The degree of meiosis I sister chromatid segregation is similar in *GAL-CLB3* and *CUP-CLB3* cells. This is discussed extensively in the revised manuscript.

Second, one must ask whether there is a method other than expressing *CLB3* from a heterologous promoter to determine why so many mechanisms, namely transcriptional and translational inhibition, are in place to keep Clb3-CDKs low during meiosis I. To our knowledge, expression from a heterologous promoter is the only way to effectively override such control mechanisms and examine the consequences of loss of these controls.

*A) For example, with respect to the physiological role of outer-kinetochore disassembly during prophase, the authors' interpretation is that this process is redundant with suppression of cyclins in preventing premature kinetochore-spindle connections. But it seems that prevention of prophase spindle assembly completely overrides any need to disassemble the outer kinetochore; bipolar spindles never form in prophase I (Figure 1)*.

Of course, without a spindle chromosomes cannot segregate. As spindle formation is a prerequisite for the observed phenotype, Clb-CDK expression is an absolute must. However three observations indicate that both, a cyclin-CDK induced bipolar spindle and a functional kinetochore must be present during prophase I in order to cause premature microtubule-kinetochore interactions that ultimately result in meiosis I chromosome segregation defects.

First, *CLB3*-induced meiosis I chromosome segregation defects are completely dependent on a functional kinetochore during prophase I (Figure 5 and Figure 5 – Figure Supplement 5). Second, whereas early *CLB3* misexpression causes defects in meiosis I chromosome segregation, *CLB3* misexpression during late prophase I has no effect on chromosome segregation (Figure 6A). This difference is likely due to the outer kinetochore being present during premeiotic S phase and early prophase I but not late prophase I (Figure 6E). Most importantly, during late prophase I, only both, *CLB3* and outer kinetochore component expression, are able to induce meiosis I sister chromatid segregation (Figure 8G, Figure 9). This result demonstrates that the presence of cyclin alone during late prophase I is not sufficient to cause sister chromatid segregation during meiosis I, and that both pathways must be repressed during prophase I to establish the meiosis I-specific chromosome segregation pattern.

These arguments have been added to the discussion to ensure that the reader understands the contribution of Clb-CDK induced premature spindle formation and outer kinetochore assembly to the observed phenotype.

*B) Moreover, artificial assembly of the outer kinetochore in prophase I (CUP-NDC80 CUP-HSK3) does not influence MI segregation unless CLB3 is also artificially expressed. This is an important paradox that must be addressed*.

Ectopic expression of Ndc80 and Hsk3 alone does not affect meiosis I chromosome segregation, but neither does *CUP-CLB3* when expressed only during late prophase I. The only point we are trying to make here is that in addition to formation of the spindle early in prophase I, a functional kinetochore must be present to observe a phenotype. Our data clearly show that outer kinetochores are disassembled during prophase I and that this disassembly precludes *CUP-CLB3* from interfering with meiosis I chromosome segregation. As mentioned above, we extended the discussion of these points to make this conclusion clearer.

*It remains completely unproven whether the physiological role of outer kinetochore disassembly in prophase I is to prevent premature spindle-kinetochore connections. It seems equally possible that SPB-kinetochore connections must be dissociated for normal execution of prophase I events, e.g. telomere-nuclear envelope attachment, chromosome movement, etc*.

It is entirely possible that disassembly of the outer kinetochore serves additional purposes. This possibility is now mentioned in the Discussion.

*5. For most experiments, CUP-CLB3 is induced so early that cells may still be in S phase. FACs analysis for representative time courses should be shown. The induction of Ndt80 expression at 4.5 hrs is also questionably early - has a majority of the culture reached the ndt80 arrest point at 4.5 hrs? The concern is that early, non-physiological expression of Cdc5 is also occurring in these experiments and may actually be required for the observed phenomena. What happens if Ndt80 is expressed later? There isn't a single time-course in this paper to demonstrate when MI occurs in a typical wild-type culture. The time-course in FigureS1E shows that crossovers don't form until 7 hrs making it unlikely that cells were in mid/late pachytene at 4.5 hrs*.

New Figure 1E shows progression of wild-type cells through sporulation. By 2:15 hours, 43.2 percent of cells have replicated their DNA, by 4.5 hours all cells are in G2. Using very similar conditions a recent publication by Blitzblau et al. (PLOS Genetics 2012) has reported even faster kinetics with bulk DNA replication being completed by 3 hours. Sporulation kinetics vary significantly between experiments and in earlier experiments we observed divisions as early as 4 hrs in an unperturbed meiosis. These fast kinetics are most likely due to the fact that most of the strains we used for these studies were prototrophs. It should also be noted that the strains used in old Figure S1E (new Figure 2 – Figure Supplement 5) are not prototrophic which results in slower meiotic kinetics. However the reviewers are correct in that not all cells have replicated their DNA yet by 2:15 hours. Thus we modified the text to indicate that the bulk of cells are in S phase or early prophase I.

The reviewers were also concerned that premature expression of *NDT80* and its target *CDC5*, 4.5 hours after transfer into sporulation medium could contribute to the observed phenotypes. We have addressed this concern by examining chromosome segregation when *NDT80* was expressed substantially later, 6, 7 or 8 hours after induction of sporulation. At these times all cells had arrested in pachytene as judged by FACS analysis. The data in new Figure 9 show that concomitant expression of *NDT80* from the *GAL1-10* promoter and *CLB3*, *NDC80* and *HSK3* from the *CUP1* promoter after 6, 7, or 8 hours in sporulation medium leads to a similar phenotype as expression of the genes after 4.5 hours in sporulation medium. Thus, expression of *NDT80* early during meiosis, 4.5 hours after transfer into sporulation medium, is not responsible for the observed phenotypes. Furthermore, meiosis I sister chromatid segregation is also seen in cells lacking the *NDT80* block release system (Carlile and Amon, 2008, Figure 1C).

*6. Similarly it is an overstatement to write categorically that the work shows two independent mechanisms for timely engagement of the microtubules with the kinetochore. Manipulating the synthesis of the outer kinetochore proteins enhances the Cdk effects, but perhaps this is because the unknown kinase targets are not completely modified? One might suggest that the transcription or translation of the kinetochore proteins is regulated by a repressor whose activity is ablated by Cdk function*.

The reviewers’ assessment is only accurate when Clb3-CDKs are expressed during premeiotic S phase/early prophase I. As mentioned above, when the outer kinetochore is fully disassembled, by late prophase I, Clb3-CDK misexpression alone is insufficient to cause MI sister chromatid segregation (Figure 6A, 8G, and Figure 9).

A scenario where outer kinetochore assembly is under Clb-CDK control is possible, but unlikely. Such a mechanism is not expected to be sensitive to when Clb3-CDKs are expressed during prophase I.